# Phytochemical Analysis of the Extract from Berries of *Schisandra chinensis* Turcz. (Baill.) and Its Anti-Platelet Potential In Vitro

**DOI:** 10.3390/ijms26030984

**Published:** 2025-01-24

**Authors:** Natalia Sławińska, Bogdan Kontek, Jerzy Żuchowski, Barbara Moniuszko-Szajwaj, Jacek Białecki, Kamil Zakrzewski, Paulina Bogusz, Anna Stochmal, Beata Olas

**Affiliations:** 1Department of General Biochemistry, Faculty of Biology and Environmental Protection, University of Lodz, 90-236 Lodz, Poland; natalia.slawinska@edu.uni.lodz.pl (N.S.); bogdan.kontek@biol.uni.lodz.pl (B.K.); kamil.zakrzewski@edu.uni.lodz.pl (K.Z.); 2Department of Biochemistry and Crop Quality, Institute of Soil Science and Plant Cultivation–State Research Institute, 24-100 Pulawy, Poland; jzuchowski@iung.pulawy.pl (J.Ż.); bszajwaj@iung.pulawy.pl (B.M.-S.); asf@iung.pulawy.pl (A.S.); 3Fertilizer Research Group, Lukasiewicz Research Network–New Chemical Synthesis Institute, 24-110 Pulawy, Poland; paulina.bogusz@ins.lukasiewicz.gov.pl

**Keywords:** blood platelet, coagulation, hemostasis, *Schisandra chinensis*

## Abstract

*Schisandra chinensis* Turcz. (Baill.) is a dioecious vine belonging to the Schisandraceae family. Its berries show beneficial activities, including cardioprotective, antioxidant, and anti-inflammatory. We examined the chemical content of *S. chinensis* berry extract and its antiplatelet potential in vitro. The antiplatelet activity assays included measurements of thrombus formation in full blood (with Total Thrombus-formation Analysis System) and platelet activation and adhesion. We also assessed the extract’s effect on coagulation times in human plasma and its cytotoxicity toward blood platelets based on extracellular lactate dehydrogenase activity. The most important constituents of the extract were dibenzocyclooctadiene lignans; schisandrin was the dominant compound. *S. chinensis* berry extract at the concentration of 50 μg/mL inhibited thrombus formation by approximately 15%. The adhesion of unstimulated and thrombin-activated blood platelets to collagen was inhibited by all used concentrations of the extract (0.5–50 μg/mL), while the adhesion of adenosine diphosphate (ADP)-activated platelets to fibrinogen was inhibited only by the concentrations of 10 and 50 μg/mL. The extract also inhibited the exposition of the active form of GPIIb/IIIa on the surface of platelets stimulated with 10 μM ADP (at 0.5–50 μg/mL) and 20 μM ADP (at 50 μg/mL). The exposition of P-selectin was inhibited only by the extract at the concentrations of 5–50 μg/mL in platelets stimulated with 10 μg/mL collagen. Moreover, the extract was not cytotoxic toward blood platelets. This indicates that *S. chinensis* berries hold promise as new antiplatelet agents, but more studies are needed to determine their mechanisms of action and in vivo efficiency.

## 1. Introduction

*Schisandra chinensis* Turcz. (Baill.) is a dioecious vine belonging to the Schisandraceae family. It occurs naturally in the forests of southeastern Siberia (Primorski, Amurski, Khabarovski regions, Sakhalin), northeastern China, Korea, and Japan [1,2]. Recently, Valíčková et al. [3] have shown that *S. chinensis* has the potential to be used in the production of nutrient supplements due to its adaptogenic properties. The primary property of all adaptogens is to eliminate the effects of stress and help the body adapt to unfavorable external conditions [3].

*S. chinensis* fruit, also called the “fruit of five flavors” (wǔwèizǐ; named after its sweet peel, sour pulp, bitter and astringent seeds, and salty grain) has been used in Chinese medicine to treat different disorders and diseases for a long time. Indigenous peoples of the Far Eastern regions of Russia used it to reduce hunger, thirst, and exhaustion [1,4]. In traditional Chinese medicine, *S. chinensis* is often used in the treatment of palpitation [5]; the 2020-year edition of Chinese Pharmacopoeia contains 100 prescriptions containing *S. chinensis* [6]. Studies have shown that *S. chinensis* berries have cardioprotective potential [7,8] and can reduce oxidative stress, obesity, and inflammation [5,9,10]. Due to the medicinal and health-promoting properties of this plant, many food supplements made from *S. chinensis* fruits are available on the market.

Different components of *S. chinensis*, especially lignans (including schisandrin, also called schizandrin, schisandrol A, or wǔwèizǐ sù A), could play a significant role in the prophylaxis and treatment of cardiovascular diseases (CVDs), for example, in myocardial infarction and hypertension [11,12,13,14,15]. Kim et al. [16] have noted that the extract, fractions, and lignans (pregomisin and gomisin N) from *S. chinensis* fruits inhibited arachidonic acid-induced aggregation of rabbit washed platelets [16]. Kim et al. [17] also evaluated the antiplatelet effect of combined *S. chinensis* and *Morus alba* leaves extracts in rats [17], while Jung et al. [18] and Lee et al. [19] reported that lignans from *S. chinensis* are antagonists of platelet activating factor (PAF) in rabbit washed platelets [18,19]. However, there are not enough studies that have investigated the effect of *S. chinensis* berries on various elements of hemostasis in human blood [20,21].

Hemostasis is a complex process depending on many interconnecting factors, among which blood platelets play a key role [22,23]. They can activate due to contact with various agonists, like thrombin, adenosine diphosphate (ADP), thromboxane A_2_, or collagen [24,25,26]. Upon activation, platelets change their shape and expose various proteins that aid in the coagulation process. One of those proteins is P-selectin, which is located in α granules in resting platelets. After activation, α granules fuse with the cell membrane, which exposes P-selectin on the surface of platelets, allowing for their adhesion to leukocytes and/or endothelial cells. Because P-selectin is exposed only on the surface of activated platelets, it is often used as an activation marker [23,26]. GPIIb/IIIa (integrin α_IIb_β_3_) is another protein that plays a vital role in hemostasis. In resting platelets, GPIIb/IIIa is exposed on the cell membrane in its low affinity state and transforms into a high affinity (active) form upon platelet activation [23,24,25,26]. Currently, there are three GP IIb/IIIa inhibitors (tirofiban, eptifibatide, and abciximab) used as anti-platelet drugs [27,28]. Platelet adhesion is mediated mainly by GPIb-IX-V (which binds to von Willebrand factor (vWF), which in turn binds to collagen) and GPVI, GPIa/IIa, or GPIV, which bind to collagen directly [23,24,25,26].

Cardiovascular diseases are a leading cause of death worldwide. An imbalance in hemostasis which leads to excessive clotting is a major issue in many CVDs and can lead to severe conditions, like myocardial infarction or stroke [22,23]. Increased platelet activation plays a major role in the pathology of many CVDs and is associated with adverse prognosis. To prevent this, many cardiovascular patients must take anti-platelet medication, among which acetylsalicylic acid (aspirin) is used most often. Aspirin inhibits cyclooxygenase (COX) activity, decreasing the production of thromboxane A_2_ [23]. Unfortunately, aspirin and many other anti-platelet drugs can cause adverse effects, including bleeding from the gastrointestinal tract. For this reason, researchers are searching for new anti-platelet compounds with a lower risk of side effects [24,29].

The present study examined the anti-platelet activity of the extract from *S. chinensis* berries in human whole blood and washed human blood platelets in vitro. Using washed platelets, we studied the effect of the extract on platelet adhesion to two adhesive proteins—collagen and fibrinogen. The anti-platelet activity of the extract in whole blood was evaluated with the Total Thrombus-formation Analysis System (T-TAS) and flow cytometry. We also investigated the effect of *S. chinensis* berry extract on the coagulation times of plasma in vitro (thrombin time (TT), prothrombin time (PT), and activated partial thromboplastin time (APTT)) and evaluated its cytotoxicity toward blood platelets by measuring extracellular lactate dehydrogenase (LDH) activity. The properties of the extract from *S. chinensis* berries were compared to those of the extract from sea buckthorn (*Hippophae rhamnoides* L.) berries, and a commercial product—Aronox (*Aronia melanocarpa* berry extract), which have documented anti-platelet properties and modulate other elements of hemostasis [30,31,32,33,34,35,36,37,38,39,40].

## 2. Results

### 2.1. Chemical Characteristic of the Extract from S. chinensis Berries

The composition of the *S. chinensis* fruit extract was analyzed by LC-HRMS. The extract contained many lignans, which seemed to be its major phenolic constituents (and more generally, main specialized metabolites) (Table 1; Figure 1). Among them, schisandrin (tentatively identified) was the dominant compound. The red color of the extract can be attributed to the presence of anthocyanin, cyanidin-Pen-Hex-dHex. *S. chinensis* fruits also contained small amounts of (epi)catechin, dimeric and trimeric B-type proanthocyanidins, and flavonoids (quercetin hexoside-deoxyhexoside and quercetin hexoside). Apart from phenolics, small amounts of highly oxygenated nortriterpenoids (mainly), bisnortriterpenoids, triterpenoids, and homotriterpenoids were also detected (Table 1). Moreover, the extract contained very high amounts of diverse, highly polar constituents (Figure 1).

Results of the quantitative analysis of phenolic compounds are shown in Table 2. As mentioned above, lignans were dominant phenolic compounds of the extract from *S. chinensis* fruits. The total lignan content was 29.61 ± 0.18 mg·g^−1^ of the extract (expressed as schisandrin equivalent). The total flavonoid content was 1.64 ± 0.03 mg·g^−1^ (expressed as rutin equivalent), and the content of anthocyanidins was 17.08 ± 0.35 mg·g^−1^ (expressed as cyanidin equivalent).

### 2.2. Effect of the Extract from S. chinensis Berries on the Adhesion of Washed Blood Platelets to Collagen and Fibrinogen

The anti-adhesive activity of the extract was studied using human washed blood platelets. The results were presented as percent of adhesion of the control samples. As shown in Figure 2A,B, the level of adhesion of unstimulated blood platelets and adhesion of thrombin-activated platelets was significantly reduced in the presence of all used concentrations of the extract from *S. chinensis* berries (0.5–50 µg/mL). For example, at the highest concentration (50 µg/mL), the extract inhibited the adhesion of thrombin-activated blood platelets by about 30% in comparison with control (Figure 2B). On the other hand, none of the tested concentrations of the extract (0.5–50 µg/mL) demonstrated anti-adhesive properties when adhesion of thrombin-activated platelets to fibrinogen was measured (Figure 2C). As for the ADP-activated blood platelets, three concentrations of the extract from *S. chinensis* berries (0.5, 1, and 5 µg/mL) did not have anti-adhesive activity (Figure 2D); reduced adhesion was only observed for the two highest concentrations (10 and 50 µg/mL) (Figure 2D).

### 2.3. Effect of the Extract from S. chinensis Berries on Parameters of Blood Platelet Activation Measured with Flow Cytometry in Whole Blood

Figure 3 and Figure 4 demonstrate the parameters of activation of platelets stimulated with 10 and 20 µM ADP, and 10 µg/mL collagen (measured with flow cytometry). Changes in platelet activation were noted in whole blood treated with the extract from *S. chinensis* berries at all tested concentrations (0.5–50 µg/mL), but these changes were not always statistically significant (Figure 3 and Figure 4).

The extract from *S. chinensis* berries (at all used concentrations: 0.5–50 µg/mL) significantly inhibited the exposition of the active form of GPIIb/IIIa on the surface of blood platelets stimulated with 10 µM ADP (Figure 3A). In blood platelets stimulated with 20 µM ADP, statistically significant reduction of the exposition of the active form of GPIIb/IIIa was observed only for the highest concentration of the extract (50 µg/mL) (Figure 3B). On the other hand, none of the concentrations of the extract (0.5–50 µg/mL) significantly impacted the exposition of GPIIb/IIIa on platelets stimulated by 10 µg/mL collagen, and the exposition of P-selectin on the surface of blood platelets stimulated by 10 and 20 µM ADP (Figure 3C and Figure 4A,B). However, in platelets activated by collagen, significant inhibition of the exposition of P-selectin was observed for the three highest concentrations of the extract (5, 10, and 50 µg/mL) (Figure 4C).

Moreover, none of the used concentrations (0.5–50 µg/mL) of the extract from *S. chinensis* berries significantly impacted platelet-derived microparticle formation and the exposition of GPIIb/IIIa and P-selectin on unstimulated platelets (data not presented).

### 2.4. Effect of the Extract from S. chinensis Berries on Platelet Plug Formation with T-TAS (PL-Chip) in Whole Blood

The extract from *S. chinensis* berries at the highest concentration (50 µg/mL) significantly decreased the AUC_10_ value measured by T-TAS in whole blood, indicating antiplatelet activity. The AUC_10_ parameter was decreased by approximately 15% (Figure 5A). Figure 5B shows changes in pressure that were recorded within the PL-chip in each sample at the highest concentration of the extract (50 μg/mL). The samples differed between each other; however, in most samples, the antithrombotic activity of the extract took effect in the later stages of the test. At first, the pressure was similar in the control and the extract-containing sample; after a while, maximum pressure was recorded in the control sample, while the extract prolonged the time needed to reach total occlusion.

### 2.5. Effect of the Extract from S. chinensis Berries on Coagulation Times (PT, TT, and APTT) of Plasma

The analysis of the effect of the extract from *S. chinensis* berries on the coagulation times of human plasma showed that none of the used concentrations of the extract (0.5–50 µg/mL) changed APTT, PT, and TT (*n* = 6) (data not demonstrated).

### 2.6. Effect of the Extract from S. chinensis Berries on a Marker of Blood Platelet Damage

LDH activity was measured to determine the toxic effect of the extract on blood platelets. The results demonstrate no significant difference in platelet viability after exposure to *S. chinensis* berry extract at all used concentrations (0.5–50 µg/mL) (Figure 6).

Table 3 compares the inhibitory effects of the extract from *S. chinensis* berries, the extract from sea buckthorn berries, and the extract from aronia berries (as a positive control) (50 µg/mL) on selected markers of blood platelet activation. The strongest anti-platelet activity was demonstrated by the extract from *S. chinensis* berries and the extract from aronia berries. The anti-platelet potential was determined for six markers.

## 3. Discussion

The LC-HRMS analysis showed that *S. chinensis* berry extract contained diverse lignans, which were its most important phenolic constituents. Most of these tentatively identified compounds were dibenzocyclooctadiene lignans, except for pregomisin, which belongs to the dibenzylbutane type. The presence of dibenzocyclooctadiene lignans is a characteristic feature of the Schisandraceae family; they are regarded as the main bioactive constituents of *S. chinensis* [46]. Schisandrin was the dominant compound; putative schisandrol B, angelogomisin H, deoxyschisandrin, and schisandrin B were other major lignans. These results are mostly in line with other studies [43,44,45,47]. The red color of *S. chinensis* berries is caused by anthocyanin, cyanidin-Pen-Hex-dHex. According to the literature data, this compound is most probably cyanidin 3-*O*-xylosylrutinoside [41,48]. Flavonoids, represented by rutin and quercetin hexoside, occurred in small quantities. These results are supported by the work of Mocan et al. [49], who found small amounts of rutin, quercetin 3-*O*-glucoside, and quercetin 3-*O*-galactoside in *S. chinensis* berries. Apart from phenolics, the extract also contained numerous terpenoid compounds, mainly highly oxygenated nortriterpenoids, as well as bisnortriterpenoids, triterpenoids and homotriterpenoids, all present in small amounts. Substances with identical or similar molecular formulas were previously isolated from the aerial parts or fruits of *S. chinensis* [42,46,50,51]. One of the detected bisnortriterpenoids was tentatively identified (on the basis of its determined formula) as wuweizidilactone H; this compound was earlier purified from berries of *S. chinensis* by other research groups [42,52].

The most important and new aspect of our findings is the confirmation of the anti-platelet activity of the extract from *S. chinensis* berries in various in vitro models, in both washed blood platelets and whole blood. The platelet isolation protocol can strongly influence platelet response. Ideally, all assays should be performed in conditions that are as close to the natural platelet environment as possible; however, some protocols require the separation of platelets from other blood components [53]. Using both methods allows for a broader understanding of platelet function. Here, the extract from *S. chinensis* showed antiplatelet activity in both methods that utilized full blood (flow cytometry and T-TAS) and washed platelets (platelet adhesion to fibrinogen and collagen). For the first time, we noted that *S. chinensis* extract (at the highest tested concentration–50 µg/mL) significantly prolonged the time of platelet plug formation in full blood. This effect was observed only at the highest concentration and was quite small; however, it is important to note that the antithrombotic activity of the extract could have been underestimated due to the character of the assay. The BAPA anticoagulant employed in the PL-chip blocks secondary hemostasis, including thrombin activity. The fact that the extract significantly inhibited the adhesion of thrombin-activated platelets to collagen at all used concentrations suggests that thrombin might be one of the targets of its antiplatelet activity. The extract also inhibited the adhesion of unstimulated platelets to collagen and ADP-stimulated platelets to fibrinogen. Moreover, all the tested concentrations (0.5–50 µg/mL) of the extract inhibited the exposition of the active form of GPIIb/IIIa in 10 µM ADP-stimulated platelets. Kim et al. also noted that combined extracts from *S. chinensis* and *Morus alba* leaves inhibited glycoprotein GPIIb/IIIa activation and granule secretion in rat washed platelets in vitro; however, in this study, the concentration of the extract was higher (400 μg/mL). The extracts also inhibited collagen-stimulated platelet aggregation measured by light transmission aggregometry. Moreover, 200 mg/kg/day of *S. chinensis* leaves extract decreased thrombus formation in rats; this activity was evaluated in an in vivo AV shunt model [17]. Chang et al. (2005) demonstrated that the extract from *S. chinensis* inhibits arachidonic acid-induced blood platelet aggregation as well. The authors suggested that the inhibition of cyclooxygenase is its primary mechanism of action [21]. However, it is important to note that it is difficult to compare the results from different laboratories, because the anti-platelet potential of *S. chinensis* berries varies according to experimental model, method, and dosage. For example, in a study by Kim et al. [17], *S. chinensis* fruit extract at a concentration of 2.5 mg/mL strongly extended PT, TT, and APTT; in contrast, much lower concentrations which were used in the present study (0.5–50 μg/mL) did not affect the coagulation times. It is important to note that PT, APTT and TT are screening assays with technical limitations, and more sensitive and comprehensive assays, e.g., thrombin generation, are needed to confirm the absence of any effect on the clotting pathways.

Various studies showed that lignans (including schisandrin, schisandrin A, B, and C) isolated from *S. chinensis* are its main active constituents and possess a variety of nutritional properties and biological activities [54,55,56]. The most active component of *S. chinensis* berry extract may be schisandrin, which could be the major determinant of its anti-platelet activity in vitro; however, not much is known about its mechanisms of action. In a study by Jung et al. [17], schisandrin A and B significantly inhibited the binding of platelet activating factor (PAF) to rabbit blood platelets [18]. This could explain the antiplatelet activity of *S. chinensis* extract; however, more studies that explore its potential mechanisms of action are needed to fully understand its properties. Two other compounds from *S. chinensis* fruits (pregomisin and chamigrenal) also showed antagonistic activity toward PAF, though their activity was relatively weak [57]. Other in vitro and in vivo models have also demonstrated that schisandrin has cardioprotective properties [58,59]. For example, Gong et al. [50] observed its cardioprotective activity in a myocardial ischemia/reperfusion injury murine model (in vivo) and H9c2 cardiomyocyte cell line subjected to hypoxia/reoxygenation injury (in vitro) [59]. Zhang et al. [49] also found that schisandrin promotes the recovery of myocardial tissues by enhancing cell viability and migration [58]. Recently, more information about the protective effect of schisandrin on the cardiovascular system has been described in a review paper by Wang et al. [55]. However, it did not cover the effect of this compound on blood platelets.

The role of GPIIb/IIIa is to facilitate platelet aggregation through fibrinogen binding; ADP mediates this process through platelet activation [25]. The extract from *S. chinensis* inhibited both ADP-induced exposition of GPIIb/IIIa on the surface of platelets and adhesion of ADP-stimulated (but not thrombin-stimulated) platelets to fibrinogen. Interestingly, ADP stimulation had no effect on the exposition of P-selectin on the platelet surface. This suggests that *S. chinensis* might interfere with the process of ADP-induced change of conformation of GPIIb/IIIa from its low-affinity to high-affinity state or fibrinogen binding. This could be achieved through interaction with different signaling molecules. For example, Ginsenoside-Rp3 (a Ginseng saponin) decreased ADP-induced platelet aggregation, fibrinogen binding, and fibronectin adhesion to GPIIb/IIIa through the inhibition of Src family kinases (SFKs), Src-dependent phospholipase Cγ2 (PLCγ2), and phosphatidylinositol 3-kinase/Akt (PI3K/Akt) signaling pathways [60].

It is important to evaluate the bioavailability and toxicity of various plant preparations that are to be used in drugs or supplements. The concentrations used in this study (0.5–50 μg/mL) are generally attainable upon oral administration of plant extracts to humans [61,62,63]. Only the highest concentration of the extract (50 μg/mL) had a statistically significant effect on the exposition of the active form of GPIIb/IIIa and thrombus formation in full blood; however, in most of the other assays, lower concentrations (0.5–10 μg/mL) had significant antiplatelet effects as well. None of the used concentrations of the extract from *S. chinensis* berries caused damage to human blood platelets. These results indicate that this extract should be safe for use as a natural supplement with anti-platelet activity. Moreover, the extract from *S. chinensis* berries had little to no toxicity toward various animals, including mice, rats, pigs, and rabbits [4,6]. For example, the results of Chen et al. [55] demonstrated that it did not have toxic effects in an atherosclerosis rat model [64].

Researchers have incorporated schisandrin as a key ingredient in various formulations, including tablets, capsules, liquids, and injections, to investigate its efficacy. Interestingly, schisandrin exhibited a good level of absorption. Hydroxylation and demethylation pathways are its main metabolic modifications [55].

Other active components that we identified in *S. chinensis* berries are triterpenoids, which display a wide range of biological activities, including antitumor, antiviral, hepatoprotective, neuroprotective, and anti-inflammatory properties [54]. Other studies indicate that triterpenoids and their derivatives present in fractions and extracts from various organs of sea buckthorn have anti-platelet activity [32,65].

Another group of compounds that can significantly affect the circulatory system, including blood platelets, are phenolic compounds. They are present in large quantities in many plants, including sea buckthorn and aronia [8,30,31,32,66,67]. Some of the mechanisms of their anti-platelet activity are cyclooxygenase inhibition and blocking the binding of surface receptors to adhesion proteins. An additional advantage of supplementation with phenolic compounds is the lack of side effects [66,68], but this matter needs further studies.

To conclude, the results indicate that *S. chinensis* berries could be used to produce natural nutrient supplements with cardioprotective potential, including anti-platelet activity. Nonetheless, the activity of *S. chinensis* still lacks validation in clinical settings. Future investigations into this extract and its chemical components, including schisandrin, should prioritize the following areas: (1) further understanding of the molecular mechanisms underlying its anti-platelet action, and the identification of specific targets; (2) bridging the gaps in knowledge about its in vivo efficiency by using animal models and performing clinical studies on healthy people and patients with different CVDs.

## 4. Materials and Methods

### 4.1. Reagents

Methanol (HPLC grade) was purchased from Fisher Chemical (Loughborough, United Kingdom), heptane (for chromatography), and acetonitrile (LC-MS grade) were obtained from Merck (Darmstadt, Germany). Cyanidin and schisandrin were from Merck, and rutin from PhytoLab (Vestenbergsgreuth, Germany). Monoclonal antibodies for flow cytometry (CD61-PerCP, CD62P-PE, and PAC-1-FITC) were purchased from Becton Dickinson (New Jersey, NJ, USA). Phosphate buffered saline (PBS), fibrinogen, collagen, bovine serum albumin (BSA), tris(hydroxymethyl)aminomethane (Tris), 4-nitrophenyl phosphate, adenosine diphosphate (ADP), nicotinamide adenine dinucleotide (NADH), and thrombin were purchased from Merck (Darmstadt, Germany). NaCl, sodium citrate, and citric acid were purchased from POCH (Avantor performance materials, Gliwice, Poland). Reagents needed for coagulation times measurements were purchased from Kselmed (Grudziądz, Poland). All materials and chemicals for T-TAS were purchased from Bionicum Sp. z o.o. (Warsaw, Poland). Other reagents were purchased from commercial distributors and were of the highest available grade.

### 4.2. Plant Material

Crimson-red, ripe *S. chinensis* berries, growing in spike clusters on vines, were collected in mid-September 2022 in the park and palace settlement “Lower Garden” of the Czartoryski family in Pulawy (51°24′47.698′′ N21° 57′ 20.626′′ E). The plants were planted about 40 years ago and identified by the park’s curator, Dr. Adam Wołk. After harvesting, the berries were frozen at −18 °C, and subsequently lyophilized (Gamma 2-16 LSC, Christ, Osterode am Harz, Germany). A voucher specimen marked 1/09/2022 is located at the Institute of Soil Science and Plant Cultivation (Pulawy, Poland).

### 4.3. Preparation of the Extract from S. chinensis Berries

The freeze-dried fruit (100 g) was soaked overnight in 80% methanol (*v*/*v*; 1.5 L) at ambient temperature, roughly homogenized using a kitchen blender, and subjected to a 15 min sonication in a large ultrasonic cleaning bath (Sonic-33; Polsonic, Warsaw, Poland) at ambient temperature. The collected extract was centrifuged at 3077× *g* for 15 min (3–16 KL, Sigma, Osterode am Harz, Germany) and filtered using a glass filter funnel. The extraction procedure was repeated twice, with new portions of the solvent. The pooled extracts were concentrated in a rotary evaporator (Hei-Vap, Heidolph Instruments GmbH, Schwabach, Germany) and defatted by liquid-liquid extraction with heptane. The defatted extract was rotary evaporated to remove organic solvents and reduce its volume, and freeze-dried. The extract (74.57 g) was subsequently cleaned up by SPE to remove organic acids, sugars, and other highly polar constituents. It was dissolved in 0.1% formic acid (*v*/*v*) in MilliQ water (18.2 MΩ cm) and loaded onto a short C_18_ column (60 × 50 mm; Cosmosil 140C18-Prep, 140 μm). The column was washed with 0.1% formic acid, and the bound compounds were eluted with methanol. The eluate was rotary-evaporated, and the residue was freeze-dried to yield 7.61 g of the purified extract.

### 4.4. Phytochemical Analysis of the Extract from S. chinensis Berries

The composition of *S. chinensis* fruits was analyzed by LC-HRMS, using a Thermo Ultimate 3000RS (Thermo Fisher Scientific, Waltham, MA, USA) UHPLC system equipped with a corona-charged aerosol detector (CAD) and coupled with a Bruker Impact II HD quadrupole-time of flight (Q-TOF) mass spectrometer (Bruker Daltonics GmbH, Bremen, Germany). Samples were chromatographed on an ACQUITY UPLC HSS T3 column (2.1 × 150 mm, 1.8 µm; Waters Inc., Milford, MA, USA), equipped with a pre-column, and maintained at 40 °C. The injection volume was 2 µL. The mobile phase A was 0.1% formic acid (*v*/*v*) in MilliQ water (18.2 MΩ cm), and the mobile phase B was acetonitrile containing 0.1% of formic acid (*v*/*v*). The flow rate was 0.400 mL min^−1^. The separation program started from a short isocratic elution with 5% B (0.5 min), followed by a 31 min gradient from 5 to 95% of solvent B. The elution was continued at 95% B for 7 min, then the mobile phase composition returned to the initial conditions and the column was equilibrated with 5% B for 5 min. MS analyses were performed in the negative and positive ion mode. MS settings for negative ion mode: capillary voltage 3 kV; dry gas flow 6 L min^−1^; dry gas temperature 200 °C; nebulizer pressure 2.5 bar; collision RF 750 Vpp; transfer time 100 µs; prepulse storage time 10 µs. Collision energy was set automatically in the range from 2.5 to 80 eV, depending on the *m*/*z* of a fragmented ion. The scanning values ranged from *m*/*z* 80 to m/z 2000. The settings for positive ion mode: capillary voltage 4 kV; dry gas flow 6 L min^−1^; dry gas temperature 200 °C; nebulizer pressure 2.5 bar; collision RF 750 Vpp; transfer time 100 µs; prepulse storage time 10 µs. Collision energy was set automatically in the range from 2.5 to 50 eV. The scanning range was *m*/*z* 100–2000.

Quantitative analyses of phenolic compounds were performed using an ACQUITY UPLC^®^ chromatographic system (Waters Inc., Milford, MA, USA), equipped with a PDA detector, and coupled with a triple quadrupole mass detector (TQD; Waters Inc.). The *S. chinensis* extract was separated on an ACQUITY UPLC BEH C_18_ (2.1 × 100 mm, 1.7 μm; Waters Inc.) column. The column temperature was 50 °C, the injection volume was 2.5 µL. Mobile phase was composed of Mill-Q water with 0.1% formic acid (solvent A) and acetonitrile with 0.1% formic acid (solvent B); the elution program was as follows: 0.0–0.5 min, 7% B; 0.5–7.0 min, linear gradient 7–25% B; 7.0–16.9 min, linear gradient 25–75% B; 17.0–18.0 min, 95% B; 18.1–20.0 min, 7% B. The TQ mass spectrometer was operated in negative and positive ion mode. The MS settings in negative ion mode were as follows: capillary voltage 2.80 kV, cone voltage 45 V, source temperature 150 °C, desolvation temperature 450 °C, cone gas flow (N_2_) 100 L h^−1^, desolvation gas (N_2_) flow 800 L h^−1^. In positive ion mode: capillary voltage 3.10 kV, cone voltage 60 V, the remaining parameters were the same as those described above. The scanning range was *m*/*z* 100–1200. The content of anthocyanins (the peak integration at λ = 475 nm; y = 232.90079x + 82.94488, R^2^ = 0.9989), flavonoids (the peak integration at λ = 350 nm; y = 442.56566x − 46.00717, R^2^ = 0.9999), and lignans (the peak integration at λ = 255 nm; y = 607.19720x + 131.25948, R^2^ = 0.9999) was calculated on the basis of calibration curves, and was expressed as cyanidin, rutin and schisandrin equivalents, respectively.

### 4.5. Preparation of Stock Solution of the Extract from S. chinensis Berries, the Extract from Sea Buckthorn Berries, and the Extract from Aronia Berries

The extract from *S. chinensis* berries and the extract from sea buckthorn berries were dissolved in 50% DMSO (*v*/*v*) (a universal solvent for many plant substances); however, the final concentration of DMSO in the tested human blood platelets, human plasma, and human blood was below 0.05% (*v*/*v*). The addition of a low concentration of DMSO to blood platelets, plasma, and blood has no effect on the hemostatic properties of blood platelets, plasma, and full blood (data not presented).

The extract from sea buckthorn berries contained predominantly various flavonol glycosides, mainly isorhamnetin glycosides and acylated glycosides. A more detailed description of the preparation of the extract from sea buckthorn berries can be found in Skalski et al. [32].

A stock solution of commercial product—Aronox (by Agropharm Ltd., Warsaw, Poland; batch No 020/2007k, *A. melanocarpa* berry extract, a known source of anthocyanins) was prepared in H_2_O at a concentration of 5 mg/mL.

### 4.6. Blood, Plasma, and Blood Platelet Samples

Whole human blood was collected at “Diagnostyka” blood collection center (Brzechwy 7A, Lodz, Poland). All donors were non-smoking, healthy volunteers who did not drink alcohol or take medication and anti-platelet supplementation for two weeks before blood collection. Peripheral blood was drawn into tubes with benzylsulfonyl-D-Arg-Pro-4-amidinobenzylamide (BAPA) (for T-TAS) or with citrate/phosphate/dextrose/adenine (CPDA) anticoagulant (for the remaining assays).

The analysis of blood samples was performed according to the guidelines of the Helsinki Declaration for Human Research. Informed consent form was signed by each participant one day prior to blood collection. Procedures were conducted with the consent of Bioethics Committee at the University of Łódź (2/KBBN-UŁ/III/2014).

Blood for the T-TAS assay was used within 2 h after collection.

Plasma for coagulation times measurements was separated from whole blood by differential centrifugation (MPW Med. Instruments, Warsaw, Poland) at 2800× *g*, for 20 min at room temperature).

Blood platelets were isolated from fresh whole blood by differential centrifugation. First, full blood collected in a tube with CDPA anticoagulant was centrifuged at 235× *g* for 12 min. Then the platelet-rich plasma was collected and centrifuged at 1020× *g* for 15 min. The pellet containing blood platelets was gently reconstituted in Barber’s buffer (a modified Tyrode’s buffer; 0.014 M Tris, 0.14 M NaCl, 5 mM glucose, pH 7.4). The required number of platelets (2.0 × 10^8^/mL) was confirmed by a measurement in a UV-Visible Helios-α spectrophotometer (Spectrophotometer UV/Vis Helios alpha; Unicam, Cambridge, UK) at 800 nm and adjusted as needed.

In every experiment, blood, plasma, or blood platelets were incubated for 30 min at 37 °C with the extract from *S. chinensis* berries at final concentrations of 0.5–50 µg/mL, the extract from sea buckthorn berries (50 μg/mL), the extract from aronia berries (50 μg/mL), or the extract vehicle.

### 4.7. Blood Platelet Adhesion

Blood platelet adhesion was measured based on the activity of an exoenzyme—acid phosphatase—in blood platelets, according to the method described by Bellavite et al. [69]. First, 96-well plates were coated with 100 μg/mL fibrinogen or 0.04 μg/mL collagen. To achieve this, 100 μL of fibrinogen or collagen was added to the wells, and incubated for 24 h at 4 °C, on an orbital shaker. Afterward, the wells were washed three times with TBS (pH 7.5), and 200 μL of 1% BSA (*w*/*v*) was added. The plates were incubated with BSA at 37 °C for 2 h. Meanwhile, the extracts were added to washed blood platelets at final concentrations of 0.5–50 μg/mL. The control sample was blood platelets with Barber’s buffer; the adhesion of this sample was assumed to be 100%. The samples were incubated for 30 min at 37 °C. BSA was removed, and the wells were washed three times with TBS (pH 7.5) with 0.1 mM CaCl_2_ and 0.1 mM MgCl_2_. The samples were added to the wells in triplicate. Agonists (1 U/mL thrombin or 30 μM ADP) or TBS were added to appropriate wells. Then, the plates were incubated for 1 h at 37 °C. Afterward, the wells were washed three times with PBS. 150 μL of 0.1 M citrate buffer (pH 5.4) with 5 mM *p*-nitrophenyl phosphate and 0.1% Triton X-100 (*v*/*v*) was added; the samples were incubated for 1 h at room temperature. Lastly, 100 μL of 2 M NaOH was added to the wells. Absorbance was read at 405 nm with a SPECTROstar Nano Microplate Reader (BMG LABTECH, Ortenberg, Germany).

### 4.8. Flow Cytometry Analysis

Platelet activation was assessed based on platelet-derived microparticle (PMP) formation, and the exposition of P-selectin (CD62P) and the active form of GPIIb/IIIa on the surface of blood platelets. To measure the exposition of P-selectin and the active form of GPIIb/IIIa, full blood was mixed with the extracts at concentrations of 0.5–50 μg/mL and incubated for 15 min at 37 °C. After 15 min, agonists were added to the samples (10 μM ADP, 20 μM ADP, or 10 μg/mL collagen), and the incubation continued for another 15 min (at 37 °C). After the incubation, the samples were diluted 10 times with PBS and incubated with antibodies (CD61-PerCP, CD62P-PE, and PAC-1-FITC); 3 μL of each antibody was incubated with 10 μL of diluted samples. The samples were stained for 30 min in the dark, at room temperature. Afterward, the samples were fixed with 1% CellFix (*v*/*v*) for 1 h at 37 °C. The samples were analyzed with an LSR II Flow Cytometer (Becton Dickinson, San Diego, CA, USA); at least 5000 CD61-PerCP-positive objects were recorded. Blood platelets were gated based on an FSC/SSC (forward scatter/side scatter) plot and positive staining with CD61-PerCP. Then, the percent of CD62-PE and PAC-1-FITC-positive platelets was calculated in each of the samples. This method was described previously by Rolnik et al. [70].

To measure the formation of platelet-derived microparticles, the samples were incubated with *S. chinensis* extract for 15 min at 37 °C. Then, 20 μg/mL collagen was added, and the incubation continued for another 15 min. The samples were diluted 10 times with PBS and stained with 2 μL of CD61-PE antibody for 30 min in the dark, at room temperature. The samples were fixed with 1% CellFix (*v*/*v*) for 1 h and analyzed with an LSR II Flow Cytometer (Becton Dickinson, San Diego, CA, USA). At least 10,000 CD61-PE-positive objects were recorded. PMPs were distinguished from CD61-PE-positive objects based on an FSC/SSC plot on a log/log scale. The analysis was carried out with Floreada software (https://floreada.io, accessed on 26 February 2024).

### 4.9. Total Thrombus-Formation Analysis System (T-TAS) (PL-Chip)

The T-TAS system can measure the thrombus formation process in semi-physiological conditions. Here, the PL-chip, which measures only primary hemostasis, was used. The T-TAS is a standardized whole blood flow chamber system for the measurement of in vitro thrombus formation in ready-to-use pre-coated chips with microcapillary flow channels. The system includes a platelet chip (PL-chip), which contains 26 microcapillaries coated with type I collagen and shear rates of 1500/s.

T-TAS (Fujimori Kogyo Co. Ltd., ZACROS, Tokyo, Japan) was performed according to the manufacturer’s instructions. For the T-TAS measurements, blood was pipetted into the Reservoir Set attached to the PL-chip’s flow path. In the PL-chip, BAPA-anticoagulated blood is pumped through collagen-coated microcapillaries with a 1500/s wall shear rate, which causes platelet activation and formation of platelet thrombi. The newly formed thrombi block the flow path, which increases the pressure inside the capillaries. The BAPA anticoagulant blocks secondary hemostasis through the inhibition of thrombin and factor Xa, so only primary hemostasis can take place.

The results were recorded as AUC_10_ (Area Under the Curve)—the area under the flow pressure curve recorded for 10 min after the start of the test. The AUC_10_ depicts the growth, intensity, and stability of thrombus formation. The experiments were carried out with T-TAS 01 PL (Fujimori Kogyo Co., Ltd., ZACROS, Tokyo, Japan) and data were depicted as % of control. Further information about this assay can be found in Hosokawa et al. [71]. First, the extracts were incubated with human full blood at final concentrations of 0.5–50 μg/mL (37 °C, 30 min). A control sample with 0.9% NaCl (*w*/*v*) instead of the extract was set up. Then, 320 μL of each sample was added to the PL-chip, and pressure was recorded for 10 min or until it reached 60 kPa (Time of Occlusion). The results were depicted as % of the control sample.

### 4.10. Measurement of Prothrombin Time, Thrombin Time, and Activated Partial Thromboplastin Time

Coagulation times were determined coagulometrically (optical method based on measurements of turbidity), according to the protocol described by Malinowska et al. [72]. The extract was incubated with human plasma at 37 °C for 30 min, at final concentrations of 0.5–50 µg/mL. The measurements were carried out on an Optic Coagulation Analyser, model K-3002 (Kselmed, Grudziadz, Poland). Each sample was measured in duplicate.

### 4.11. Activity of LDH

The toxic effect of the extract from *S. chinensis* berries on human blood platelets was analyzed by measuring the activity of lactate dehydrogenase (LDH), an enzyme released from damaged blood platelets [73]. Blood platelets were incubated with the extract at final concentrations of 0.5–50 µg/mL, at 37 °C for 30 min. Afterward, the samples were centrifuged (2500 rpm, 15 min, 25 °C) and 10 μL of the supernatant was added to the wells of a 96-well plate (in triplicate). Then, 0.1 M phosphate buffer (pH 7.4, 270 μL) and 0.25% nicotinamide adenine dinucleotide (NADH) (*w*/*v*) (10 μL) were added to each well. A blank which contained 280 μL of phosphate buffer and 10 μL of NADH was set up as well. The samples were mixed and incubated for 20 min at room temperature. Lastly, 0.25% pyruvate (*w*/*v*) (10 μL) was added to the samples. The samples were immediately mixed, and the absorbance was read at 1 min intervals for 10 min at 340 nm, with a SPECTROstar Nano Microplate Reader (BMG LABTECH, Ortenberg, Germany).

### 4.12. Statistical Analysis

Statistical analysis was performed with Statistica 10 (StatSoft 13.3, TIBCO Software Inc. Palo Alto, CA, USA). The distribution of data was checked by the Shapiro–Wilk test, and the homogeneity of variance by Levene’s test. Differences within and between groups were assessed with one-way ANOVA followed by Tukey’s test, or Kruskall–Wallis test. The results were presented as means ± SD. The results were considered significant at *p* ≤ 0.05. Dixon’s Q-test was used to eliminate uncertain data.

## Figures and Tables

**Figure 1 ijms-26-00984-f001:**
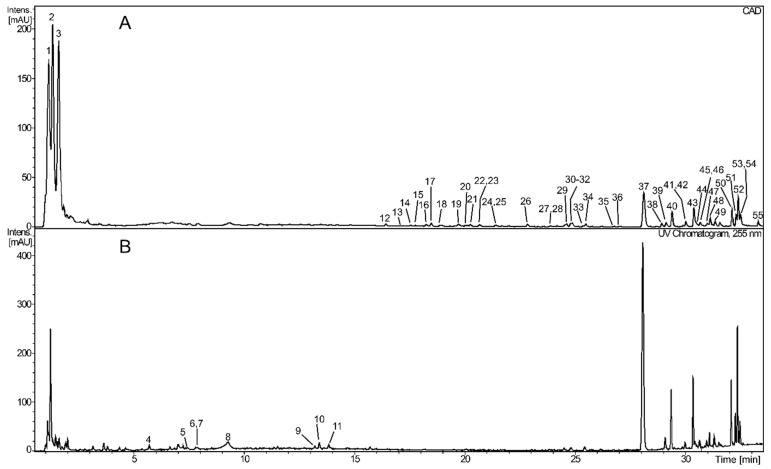
UHPLC-CAD (**A**), and UV (λ = 255 nm (**B**)) chromatograms of the extract from the fruit of *S. chinensis*. Numbers above the chromatic peaks correspond to those from Table 1.

**Figure 2 ijms-26-00984-f002:**
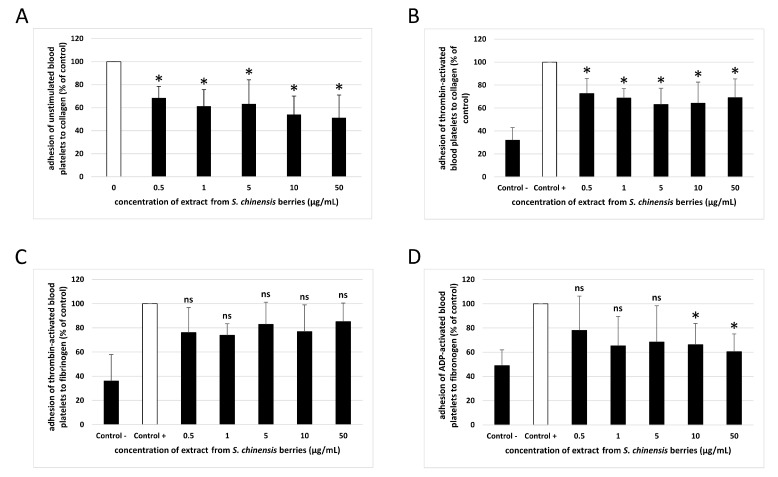
Effect of the extract from *S. chinensis* berries (at concentrations 0.5, 1, 5, 10, and 50 μg/mL) on the adhesion of unstimulated platelets to collagen (**A**), thrombin-activated platelets to collagen (**B**), thrombin-activated platelets to fibrinogen (**C**), and ADP-activated platelets to fibrinogen (**D**) (*n* = 9). In the graphs, platelet adhesion is expressed as a percentage of the ‘Control’ sample (blood platelets without the tested extract) (**A**), or the ‘Control +’ sample (platelets stimulated by agonist with no added extract) (**B**–**D**). ‘Control -‘ samples are unstimulated blood platelets. The data are expressed as means ± SD. The results were considered significant at *p* < 0.05 (* *p* < 0.05, ns—not significant). The differences between controls were significant (*p* < 0.001 (**B**,**D**), *p* < 0.01 (**C**)).

**Figure 3 ijms-26-00984-f003:**
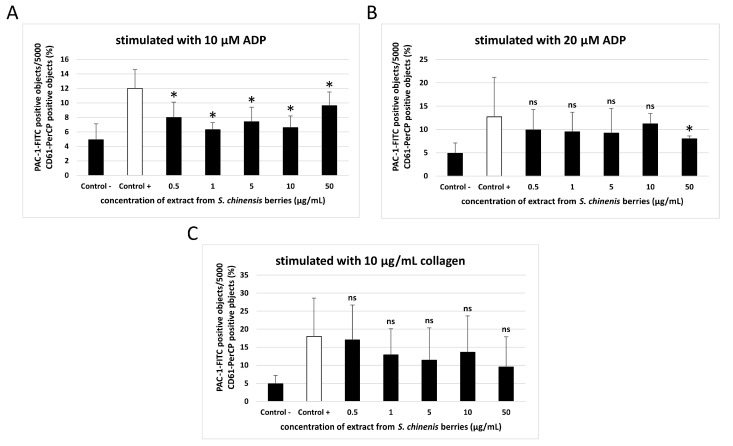
Effect of the extract from *S. chinensis* berries (at concentrations 0.5, 1, 5, 10, and 50 μg/mL) on the exposition of the active form of GPIIb/IIIa on 10 µM ADP-stimulated blood platelets (**A**), 20 µM ADP-stimulated blood platelets (**B**), and 10 µg/mL collagen-stimulated blood platelets (**C**) in whole blood samples. Blood platelets were distinguished based on the exposition of CD61. For each sample, 5000 CD61-positive objects were acquired. For the assessment of GPIIb/IIIa exposition, samples were labeled with fluorescently conjugated monoclonal antibody PAC-1/FITC. Results are shown as the percentage of platelets binding PAC-1/FITC. Data represent the means ± SD. The blood samples were drawn from 6 healthy volunteers. ‘Control -‘samples are unstimulated blood platelets, ‘Control +’ samples are platelets stimulated by agonist with no added extract. The activity of the tested extract was compared to the ‘Control +’ sample. The differences between controls were significant (*p* < 0.01 (**A**), *p* < 0.05 (**B**,**C**)). The results were considered significant at *p* < 0.05 (* *p* < 0.05, ns—not significant).

**Figure 4 ijms-26-00984-f004:**
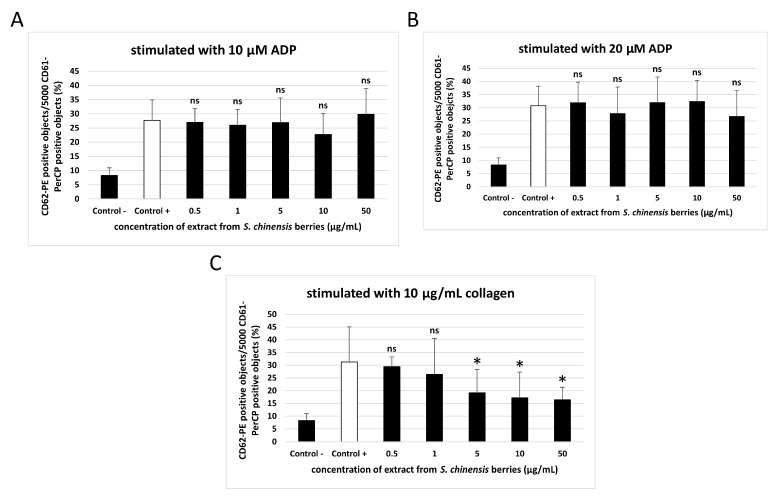
Effect of the extract from *S. chinensis* berries (at concentrations 0.5, 1, 5, 10, and 50 μg/mL) on the exposition of P-selectin on 10 µM ADP-stimulated blood platelets (**A**), 20 µM ADP-stimulated blood platelets (**B**), and 10 µg/mL collagen-stimulated blood platelets (**C**) in whole blood samples. Blood platelets were distinguished based on the exposition of CD61. For each sample, 5000 CD61-positive objects were acquired. For the assessment of P-selectin exposition, samples were labeled with fluorescently conjugated monoclonal antibody CD62P/PE. Results are shown as the percentage of platelets expressing CD62P. Data represent the means ± SD. ‘Control -‘samples are unstimulated blood platelets, ‘Control +’ samples are platelets stimulated by agonist with no added extract. The activity of the tested extract was compared to the ‘Control +’ sample. The differences between controls were significant (*p* < 0.001 (**A**,**B**), *p* < 0.05 (**C**)). The results were considered significant at *p* < 0.05 (* *p* < 0.05, ns—not significant).

**Figure 5 ijms-26-00984-f005:**
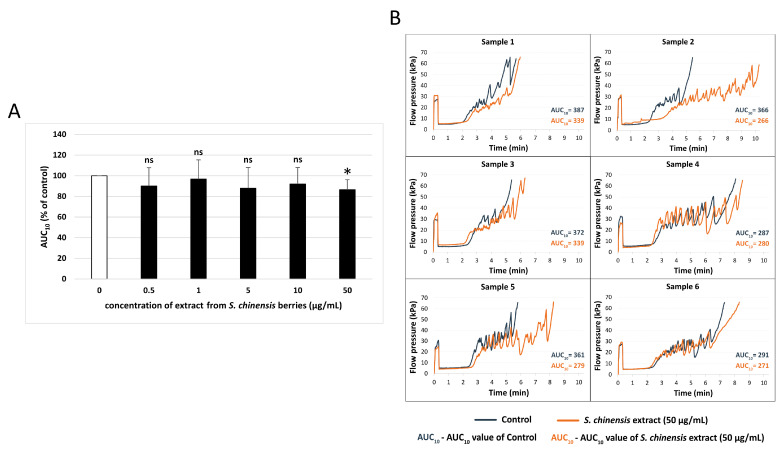
Effect of the extract from *S. chinensis* berries (at concentrations 0.5, 1, 5, 10, and 50 μg/mL) on platelet plug formation in whole blood (*n* = 6) (**A**). The samples were analyzed with T-TAS PL-chip, at the shear stress rates of 1500/s. The results are calculated as AUC_10_ (area under the curve). In the graphs, AUC_10_ is expressed as a percentage of the control sample (blood without the tested extract). The data are expressed as means ± SD. The results were considered significant at *p* < 0.05 (* *p* < 0.05, ns—not significant). (**B**) shows changes in pressure that were recorded within the PL-chip in each sample at the highest concentration of the extract (50 μg/mL).

**Figure 6 ijms-26-00984-f006:**
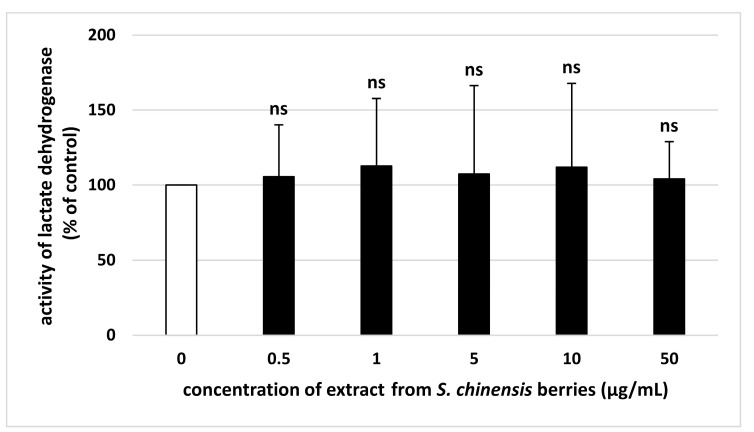
Effect of the extract from *S. chinensis* berries (at concentrations 0.5, 1, 5, 10, and 50 μg/mL) on the activity of lactate dehydrogenase (*n* = 6). The results are expressed as a percentage of the control sample. The data are expressed as means ± SD. The results were considered significant at *p* < 0.05 (ns—not significant).

**Table 1 ijms-26-00984-t001:** Specialized metabolites of the *S. chinensis* fruit extract.

No.	tR (min)	Parent Ion (*m*/*z*)	CID	Delta (ppm)	Mσ	Formula	Tentative Identification	Ref.
1–3	0.85–1.80						mixtures of polar compounds	
4	5.77	203.0823 ^-^	186.0552 (5), 159.0921 (7), 142.0641 (7), 116.0595 (17)	1.3	10.2	C_11_H_12_N_2_O_2_	tryptophan	
5	7.45	577.1352 ^-^	407.0773 (90), 289.0720 (100), 245.0821 (23)	−0.2	8.0	C_30_H_26_O_12_	(epi)C-(epi)C	
6	7.83	289.0719 ^-^	289.0720 (100), 245.0819 (73), 203.03715 (31)	−0.3	1.7	C_15_H_14_O_6_	(epi)catechin	
7	7.90	865.1982 ^-^	695.1400 (4), 525.0822 (15), 407.0785 (83), 289.0729 (100), 243.0309 (39)	0.4	15.5	C_45_H_38_O_18_	(epi)C-(epi)C-(epi)C	
8	9.30	725.1927 ^-^	339.0514 (14), 284.0329 (100)	1.0	7.2	C_32_H_39_O_19_	cyanidin-Pen-Hex-dHex	[41]
9	13.23	441.1769 ^-^	441.1769 (78), 397.1854 (28), 330.1319 (100), 217.1227 (30)	−0.6	9.3	C_21_H_30_O_10_		
10	13.45	609.1460 ^-^	300.0279 (100)	0.2	15.4	C_27_H_30_O_16_	rutin	
11	13.87	463.0882 ^-^	300.0276 (100)	0.1	13.2	C_21_H_20_O_12_	Q−3-*O*-Hex	
12	16.40	685.2927 ^-^*	639.2851 (12), 477.2348 (24), 383.1193 (100), 323.0979 (10), 263.0774 (15), 221.0664 (31), 179.0557 (28)	−0.3	8.4	C_28_H_48_O_16_	unidentified bisnorterpenoid	
13	17.09	605.2232 ^-^*	559.2182 (100), 497.2176 (65), 439.1759 (45), 351.1965 (31), 317.1391 (22)	1.3	12.2	C_29_H_36_O_11_	unidentified nortriterpenoid	
14	17.52	575.2131 ^-^	575.2138 (100), 557.2020 (20), 531.2235 (3), 455.1728 (10)	0.5	8.9	C_29_H_36_O_12_	unidentified nortriterpenoid	
15	17.74	559.2179 ^-^	559.2209 (100), 541.2087 (65), 523.2031 (50), 465.1557 (47), 447.1440 (55), 439.1785 (70), 421.1656(81)	1.1	8.0	C_29_H_36_O_11_	unidentified nortriterpenoid	
16	18.22	605.2239 ^-^*	559.2181 (100), 497.2173 (88), 479.2080 (43), 439.1762 (41), 351.1961 (42), 317.1391 (34)	0.1	7.3	C_29_H_36_O_11_	unidentified nortriterpenoid	
17	18.45	559.2189 ^-^	559.2196 (100), 541.2082 (19), 465.1531 (5), 445.1495 (6), 439.1760 (7)	−0.7	3.2	C_29_H_36_O_11_	unidentified nortriterpenoid	
18	18.85	621.2194 ^-^*	575.2142 (100), 557.2041 (28), 511.2317 (8), 455.1724 (9), 417.1558 (12)	−0.9	14.6	C_29_H_36_O_12_	unidentified nortriterpenoid	
19	19.69	605.2241^-^*	559.2188 (100), 541.2083 (14), 445.1504 (6), 347.1493 (3)	−0.3	9.1	C_29_H_36_O_11_	unidentified nortriterpenoid	
20	19.99	589.2296 ^-^*	543.2244 (47), 525.2134 (60), 507.2025(100), 481.2219 (60), 463.2128 (89), 419.2235 (94), 383.1863 (43)	−0.9	15.9	C_29_H_36_O_10_	unidentified nortriterpenoid	
21	20.24	605.2242 ^-^*	559.2184 (100), 541.2082 (40), 515.2278 (32), 497.2186 (49)	−0.4	1.6	C_29_H_36_O_11_	unidentified nortriterpenoid	
22	20.62	591.2453 ^-^*	545.2415 (2), 447.2161 (20), 411.1817 (25), 367.1917 (100), 349.1815 (22)	−1.0	11	C_29_H_38_O_10_	unidentified nortriterpenoid	
23	20.7	605.2240 ^-^*	559.2186 (100), 497.1289 (14), 457.1862 (19)	0.0	30.1	C_29_H_36_O_11_	unidentified nortriterpenoid	
24	21.38	589.2295 ^-^*	543.2245 (100), 525.2143 (46), 499.2354 (38), 481.2238 (77), 407.1863 (26)	−0.7	4.3	C_29_H_36_O_10_	unidentified nortriterpenoid	
25	21.46	593.2603 ^-^*	547.2556 (6), 529.2449 (25), 511.2336 (14), 485.2538 (16), 467.2442 (100), 427.2138 (14)	0.1	12.2	C_29_H_40_O_10_	unidentified nortriterpenoid	
26	22.83	577.2292 ^-^*	531.2238 (100), 513.2118 (11), 451.2120 (16), 297.1485 (9)	−0.3	12.8	C_28_H_36_O_10_	wuweizidilactone H/isomer	[42]
27	23.87	603.208 ^-^*	557.2037 (100), 481.1507 (17), 455.1716 (55), 437.1604 (13), 365.1393 (7)	0.5	10.4	C_29_H_34_O_11_	unidentified nortriterpenoid	
28	24.17	579.2447 ^-^*	533.2387 (100), 497.2195 (29), 475.2005 (35), 453.2270 (28), 435.2166 (24)	0.0	13.1	C_28_H_38_O_10_	unidentified bisnorterpenoid	
29	24.6	619.2757 ^-^*	573.2733 (1), 531.2596 (100), 513.2490 (16), 495.2399 (8), 211.0975 (9)	0.5	10.5	C_31_H_42_O_10_	unidentified homotriterpenoid	
30	24.79	633.2554 ^-^*	587.2515 (1), 509.2182 (29), 491.2076 (13), 465.2284 (19), 447.2180 (100), 429.2077 (21)	−0.2	6.6	C_31_H_40_O_11_	unidentified homotriterpenoid	
31	24.79	603.2088 ^-^*	557.2032 (100), 499.1618 (10), 481.1509 (18), 455.1707 (56), 437.1600 (13)	0.0	17.6	C_29_H_34_O_11_	unidentified nortriterpenoid	
32	24.89	635.2710 ^-^*	589.2654 (4), 571.2546 (21), 509.2544 (100), 491.2440 (14), 467.2435 (54), 449.2329 (34)	−0.1	11.9	C_31_H_42_O_11_	unidentified homotriterpenoid	
33	25.36	575.2476 ^+^	557.2372 (11), 497.2162 (27), 479.2057 (69), 461.1951 (33), 437.1953 (51), 419.1846 (100)	1.9	8.9	C_30_H_38_O_11_	unidentified triterpenoid	
34	25.47	587.2126 ^-^*	541.2071 (100), 483.1655 (16), 465.1552 (20), 439.1759 (34), 365.1394 (12), 215.0715 (4)	1.4	3.9	C_29_H_34_O_10_	unidentified nortriterpenoid	
35	26.75	577.2638 ^+^	559.2533 (46), 541.2434 (29), 499.2316 (49), 481.2218 (100), 463.2107 (53), 439.2112 (66), 421.2004 (95)	0.9	9.2	C_30_H_40_O_11_	unidentified triterpenoid	
36	27.02	619.2764 ^-^*	573.2706 (71), 531.2591 (100)	−0.6	8.8	C_31_H_42_O_10_	unidentified homotriterpenoid	
37	28.12	433.2218 ^+^	415.2112 (100)	0.6	10.7	C_24_H_32_O_7_	schisandrin	[43,44,45]
38	28.93	701.2813 ^-^*	531.2215 (66), 513.2110 (100), 495.2043 (55), 477.1932 (32), 451.2151 (41), 433.2003 (64), 415.1912 (66)	0.2	10.5	C_35_H_44_O_12_	unidentified	
39	29.13	389.1960 ^+^	389.1963 (100), 357.1699 (7), 319.1180 (3), 287.0920 (3)	−0.4	1.7	C_22_H_28_O_6_	gomisin J/isomer	[43,45]
40	29.40	417.1911 ^+^	399.1804 (100), 369.1694 (13)	−0.8	7.3	C_23_H_28_O_7_	schisandrol B/isomer	[43,44,45]
41	30.03	391.2116 ^+^	391.2115 (100), 237.1485 (16)	−0.2	2.1	C_22_H_30_O_6_	pregomisin/isomer	[45]
42	30.03	501.2482 ^+^	483.2379 (65), 401.1961 (100)	0.1	11.4	C_28_H_36_O_8_	angeloylgomisin H/tigloylgomisin H/isomer	[43,44,45]
43	30.38	501.2483 ^+^	483.2377 (26), 401.1960 (100)	0.1	5.5	C_28_H_36_O_8_	angeloylgomisin H/tigloylgomisin H/isomer	[43,44,45]
44	30.46	523.2321 ^+^	505.2221 (100), 401.1960 (49)	1.0	2.4	C_30_H_34_O_8_	benzoylgomisin H/isomer	[45]
45	30.67	548.2850 ^+#^	431.2063 (100)	0.7	7.9	C_29_H_38_O_9_	angeloylgomisin Q/tigloylgomisin Q/isomer	[44,45]
46	30.70	554.2387 ^+#^	415.1754 (100)	−0.5	12.2	C_30_H_32_O_9_	schisantherin A/gomisin G/isomer	[43,44,45]
47	30.99	532.2542 ^+#^	415.1758 (100)	−0.1	18.1	C_28_H_34_O_9_	schisantherin B/isomer	[43,44,45]
48	31.12	532.2542 ^+#^	415.1755 (100), 371.1491 (7)	−0.2	3.0	C_28_H_34_O_9_	gomisin F/angeloylgomisin P/tigloylgomisin P/isomer	[43,45]
49	31.32	403.2118 ^+^	403.2120 (100), 371.1857 (6)	−0.8	8.1	C_23_H_30_O_6_	schisanhenol/isomer	[43,45]
50	32.08	417.2275 ^+^		−0.8	4.7	C_24_H_32_O_6_	schisandrin A (deoxyschisandrin)/isomer	[43,44,45]
51	32.30	401.1962 ^+^	401.1964 (100), 331.1183 (5)	−0.9	2.1	C_23_H_28_O_6_	gomisin N/isomer	[43,45]
52	32.40	401.1962 ^+^	401.1965 (100), 331.1183 (5)	−0.9	5.7	C_23_H_28_O_6_	schisandrin B (γ-schisandrin)/isomer	[43,44,45]
53	32.49	385.1649 ^+^	385.1648 (100), 355.1544 (5), 315.0871 (5), 285.0765 (3)	−0.9	5.5	C_22_H_24_O_6_	schisandrin C/isomer	[43,45]
54	32.52	485.2536 ^+^	485.2538 (100), 403.2116 (32)	−0.4	7.3	C_28_H_36_O_7_	unidentified	
55	33.31	279.2326 ^-^	279.2326 (100)	1.1	14.3	C_18_H_32_O_2_	octadecadienoic acid	

^-^—negative ion mode; +—positive ion mode; *—formic acid adduct; ^#^—NH_4_^+^ adduct; C—catechin, Q—quercetin, dHex—deoxyhexose; Hex—hexose; Pen—pentose. **1**. [41]; **2.** [42]; **3**. [43]; **4**. [44]; **5**. [45].

**Table 2 ijms-26-00984-t002:** Content of major phenolic compounds in the extract from the fruit of *S. chinensis*.

tR (min)	*m*/*z* *	Identification	Content (mg·g^−1^)
2.21	727	cyanidin-Pen-Hex-dHex	17.08 ± 0.35 ^$^
4.17	611	rutin	1.12 ± 0.02
4.32	465	quercetin-Hex	0.52 ± 0.01 ^#^
11.39	433	schisandrin	23.07 ± 0.31
11.86	389	gomisin J	Traces
11.99	417	schisandrol B	1.16 ± 0.01 ^^^
12.97	501	angeloylgomisin H/tigloylgomisin H	1.61 ± 0.02 ^^^
15.81	401	schisandrin B	2.12 ± 0.11 ^^^
16.15	385	schisandrin C	0.84 ± 0.06 ^^^

*—positive ion mode; $—cyanidin equivalent; #—rutin equivalent; ^—schisandrin equivalent; Hex—hexose; dHex—deoxyhexose; Pen—pentose.

**Table 3 ijms-26-00984-t003:** A comparison of the anti-platelet activity of the extract from *S. chinensis* berries (50 µg/mL), the extract from sea buckthorn berries (50 µg/mL), and the extract from aronia berries (50 µg/mL) in washed blood platelets (measured by adhesion to adhesive proteins (collagen and fibrinogen) and in whole blood (measured by T-TAS and flow cytometry).

Inhibitory Effect of Plant Extract on Platelet Activation (%)	*S. chinensis* Berries	Sea Buckthorn Berries	Aronia Berries
Inhibition of adhesion of thrombin-activated platelet to collagen	Anti-platelet activity (32.7 ± 14.2; *p* < 0.05)	No effect (17.4 ± 16.4; *p* > 0.05)	Anti-platelet activity (67.4 ± 15.5; *p* < 0.05)
Inhibition of adhesion of thrombin-activated platelet to fibrinogen	No effect (12.2 ± 7.8; *p* > 0.05)	No effect (15.5 ± 12.2; *p* > 0.05)	No effect (12.9 ± 7.5; *p* > 0.05)
Inhibition of adhesion of ADP-activated platelet to fibrinogen	Anti-platelet activity (42.6 ± 24.2; *p* < 0.05)	Anti-platelet activity (32.3 ± 10.1; *p* < 0.05)	No inhibitory effect (only increase; 15.3 ± 7.5; *p* > 0.05)
Inhibition of platelet plug formation (measured by T-TAS)	Anti-platelet activity (13.5 ± 9.7; *p* < 0.05)	No inhibitory effect (only increase; 21.2 ± 9.9; *p* > 0.05)	No inhibitory effect (only increase; 16.9 ± 9.2; *p* > 0.05)
Inhibition of GPIIb/IIIa exposition in 10 µM ADP-activated platelets	Anti-platelet activity (23.4 ± 5.5; *p* < 0.05)	No inhibitory effect (5.7 ± 4.2; *p* > 0.05)	Anti-platelet activity (23.9 ± 4.9; *p* < 0.05)
Inhibition of GPIIb/IIIa exposition in 20 µM ADP-activated platelets	Anti-platelet activity (35.7 ± 4.6; *p* < 0.05)	No inhibitory effect (only increase; 17.7 ± 8.1; *p* > 0.05)	Anti-platelet activity (24.1 ± 6.7; *p* < 0.05)
Inhibition of GPIIb/IIIa exposition in collagen-activated platelets	No inhibitory effect (44.4 ± 14.4; *p* > 0.05)	Anti-platelet activity (24.5 ± 6.6; *p* < 0.05)	Anti-platelet activity (26.9 ± 7.8; *p* < 0.05)
Inhibition of P-selectin exposition in 10 µM ADP-activated platelets	No inhibitory effect (only increase; 12.4 ± 5.9; *p* > 0.05)	No inhibitory effect (9.9 ± 4.1; *p* > 0.05)	No inhibitory effect (only increase; 18.3 ± 11.1; *p* > 0.05)
Inhibition of P-selectin exposition in 20 µM ADP-activated platelets	No inhibitory effect (14.9 ± 10.1; *p* > 0.05)	No inhibitory effect (10.4 ± 4.5; *p* > 0.05)	Anti-platelet activity (22.8 ± 6.6; *p* < 0.05)
Inhibition of P-selectin exposition in collagen-activated platelets	Anti-platelet activity (47.2 ± 9.1; *p* < 0.05)	No inhibitory effect (12.7 ± 5.8; *p* > 0.05)	Anti-platelet activity (24.7 ± 8.8; *p* < 0.05)

## Data Availability

The datasets used and/or analyzed during the current study are available from the corresponding author on reasonable request.

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
