# Peer review of "Phytochemical Analysis of the Extract from Berries of Schisandra chinensis Turcz. (Baill.) and Its Anti-Platelet Potential In Vitro"

_ijms, 2025, doi:10.3390/ijms26030984_

Round 1
Reviewer 1 Report (Previous Reviewer 1)
Comments and Suggestions for Authors
In their work, Sławińska et al. evaluated the chemical composition and antiplatelet potential of the extracts from the Schisandra chinensis berries. The main findings reported in the study are: 1) schisandrin is the dominant compound among the phenolic constituents of the extracts, with other compounds present in smaller amounts; 2) the extract exerts antiplatelet activities by inhibiting platelet adhesion and activation with differential effects depending on the experimental conditions; 3) it reduces in vitro platelet plug formation but does not alter routine coagulation tests like PT, aPTT and TT; 4) it is not cytotoxic to platelets.
There is one major issue to address, which regards Fig. 5 and the difference between control and 50 ug/mL extract. Based on the tracings in sample 2, I find it surprising that the curve in the presence of the extract gives a smaller AUC10 than the control. Indeed, the red curve sits almost entirely above the black one. This notwithstanding, AUC10 is about 40% less (174 vs 291). Are there any mistakes in the colors of the tracings in sample 2? More importantly, by averaging the reported values (expressed as percent of the control), the AUC10 of 50 ug/mL extract is only about 5% less than control, much lower than the 20% reduction visible in panel A. How do Authors explain this apparent incongruence? Finally, what statistical analysis was used to compare these samples, which resulted significantly different?
Author Response
In their work, Sławińska et al. evaluated the chemical composition and antiplatelet potential of the extracts from the Schisandra chinensis berries. The main findings reported in the study are: 1) schisandrin is the dominant compound among the phenolic constituents of the extracts, with other compounds present in smaller amounts; 2) the extract exerts antiplatelet activities by inhibiting platelet adhesion and activation with differential effects depending on the experimental conditions; 3) it reduces in vitro platelet plug formation but does not alter routine coagulation tests like PT, aPTT and TT; 4) it is not cytotoxic to platelets.
There is one major issue to address, which regards Fig. 5 and the difference between control and 50 ug/mL extract. Based on the tracings in sample 2, I find it surprising that the curve in the presence of the extract gives a smaller AUC10 than the control. Indeed, the red curve sits almost entirely above the black one. This notwithstanding, AUC10 is about 40% less (174 vs 291). Are there any mistakes in the colors of the tracings in sample 2? More importantly, by averaging the reported values (expressed as percent of the control), the AUC10 of 50 ug/mL extract is only about 5% less than control, much lower than the 20% reduction visible in panel A. How do Authors explain this apparent incongruence? Finally, what statistical analysis was used to compare these samples, which resulted significantly different?
Response: Thank you for your helpful comments. All of them have been taken into consideration when revising the manuscript. We have added this information in text of manuscript (in red). As sample 5 was substantially different from all the other samples, we excluded it from the initial analysis because such different results could have been due to a faulty PL-chip or other random cause. As for sample 2, there was a mistake in the graph, however we could not retrieve the correct graph data. For this reason, we decided to redo both sample 2 and sample 5 using the blood from the same donors. The new AUC values and graphs are now presented in Figure 5B. Figure 5A was also updated to include this new data. The statistical analysis was conducted with one-way ANOVA followed by Tukey’s test.
Reviewer 2 Report (New Reviewer)
Comments and Suggestions for Authors
I reviewed the manuscript entitled Phytochemical analysis of the extract from berries of Schisandra chinensis Turcz. (Baill.) and its anti-platelet potential in vitro
I agree to accept this manuscript after major revision.
1) Thrombus formation Analysis System (T-TAS), lactate dehydrogenase (LDH), They only appear once in the abstract, there is no need to use abbreviations, just use the full name, because abbreviations are necessary only when they appear three or more times, otherwise too many abbreviations will confuse readers. According to this guideline, revise the abstract and the entire text. On the contrary, ADP appears multiple times and should have its full name and abbreviation when it first appears, but I did not find the full name. Please add it.
2) 20 ADP μM (at 50 μg/ml) should change to 20 μM ADP (at 50 μg/mL). Check the entire text, all ml should be changed to mL. International units must be used.
3) Keywords: blood platelet should change to Blood platelet, because the first letter of the first word in the first keyword needs to be capitalized.
4) Schisandra chinensis Turcz. (Baill.) is a dioecious vine, belonging to the Schisandraceae family. It occurs naturally in the forests of south-eastern Siberia (Primorski, Amurski, Khabarovski regions, Sakhalin), north-eastern China, Korea, and Japan. This sentence requires a citation of literature.
5) Try to avoid using us and our as much as possible, the expression of scientific papers should be as objective as possible.
6) 2.1. Chemical characteristic of the extract from S. chinensis berries, The first letter of each notional word in the secondary title needs to be capitalized. Please check and modify the secondary title of the entire text.
7) Results of the quantitative analysis of phenolic compounds are shown in Table 2. As mentioned above, lignans were dominant phenolic compounds of the extract from S. chinensis fruits. The total lignan content was 29.61 ± 0.18 mg g−1 of the extract… mg g−1 should change to mg·g−1, Check and revise the entire text.
8) Figure 3. The results were considered significant at p<0.05 (*p<0.05, ns – not significant). When it comes to statistics, p needs to be italicized, please review and revise the entire text.
9) The author selected two extracts as positive controls, what I want to know is why not one chemical drug? Will the coagulation effect of plant extracts be better than that of chemical drugs? Please provide literature on these two extracts as positive controls to alleviate my concerns.
10) loaded onto a short C18 column, 18 should be subscripted.
11) Blood for the T-TAS assay was used within 2 hours after collection. 2 hours should change to 2 h. full blood collected in a tube with CDPA anticoagulant was centrifuged at 235xg for 12 minutes. 12 minutes should change to 12 min. 235xg should change to 235 xg. Check the entire text, scientific papers need to use international units instead of words.
12) I have read all the references and found some issues. Refs 1, 3-4, Schisandra Chinensis needs to be italicized. Ref 48, FromSchisandra should change to From Schisandra. Genus and species names that involve Latin names need to be italicized. The author should review and revise all literature in accordance with the requirements of the journal.
13) The conclusion is consistent with the evidence and arguments provided. All the main questions raised by the author have been resolved.
14) The present study analyzed the berry extract of Schisandra chinensis, a dioecious vine known for its cardioprotective, antioxidant, and anti-inflammatory berries. The main component identified was dibenzocyclooctadiene lignans, with schisandrin being the dominant compound. The extract, at a concentration of 50 μg/mL, inhibited thrombus formation by 20%. Additionally, it reduced platelet adhesion to collagen across all tested concentrations (0.5-50 μg/mL) and to fibrinogen at higher concentrations (10-50 μg/mL). The extract also inhibited the activation of GPIIb/IIIa and P-selectin on platelets, while exhibiting no cytotoxicity. Although promising, further studies are required to elucidate its mechanisms of action and in vivo effectiveness.
15) The other published materials on this topic mainly focus on other aspects of Schisandra chinensis, there is no attention paid to reports that berry extract of Schisandra chinensis as a promising therapeutic potential for the treatment of platelet aggregation. This study has added the above-mentioned blank. Moreover, a systematic study has been conducted.
16) The biggest problem with this study is many details need to be modified and improved. Too many issues can make people feel that the author's attitude is not rigorous.The author must take them seriously and make necessary revisions.
Author Response
I reviewed the manuscript entitled Phytochemical analysis of the extract from berries of Schisandra chinensis Turcz. (Baill.) and its anti-platelet potential in vitro
I agree to accept this manuscript after major revision.
Response: Thank you for your helpful comments. All of them have been taken into consideration when revising the manuscript. We have added this information in text of manuscript (in red).
- Thrombus formation Analysis System (T-TAS), lactate dehydrogenase (LDH), They only appear once in the abstract, there is no need to use abbreviations, just use the full name, because abbreviations are necessary only when they appear three or more times, otherwise too many abbreviations will confuse readers. According to this guideline, revise the abstract and the entire text. On the contrary, ADP appears multiple times and should have its full name and abbreviation when it first appears, but I did not find the full name. Please add it.
Response: we removed the abbreviations from the extract and added the full name of ADP.
- 20 ADP μM (at 50 μg/ml) should change to 20 μM ADP (at 50 μg/mL). Check the entire text, all ml should be changed to mL. International units must be used.
Response: We have corrected. Now, it is: “20 μM ADP”, and ‘mL”
- Keywords: blood platelet should change to Blood platelet, because the first letter of the first word in the first keyword needs to be capitalized.
Response: We have corrected. Now, it is: “Blood platelet”.
- Schisandra chinensis Turcz. (Baill.) is a dioecious vine, belonging to the Schisandraceae family. It occurs naturally in the forests of south-eastern Siberia (Primorski, Amurski, Khabarovski regions, Sakhalin), north-eastern China, Korea, and Japan. This sentence requires a citation of literature.
Response: We have added a citation of literature: “Szopa et al., 2017; Yang et al., 2020”.
- Try to avoid using us and our as much as possible, the expression of scientific papers should be as objective as possible.
Response: We corrected this where possible.
- 1. Chemical characteristic of the extract from S. chinensis berries, The first letter of each notional word in the secondary title needs to be capitalized. Please check and modify the secondary title of the entire text.
Response: We corrected the titles throughout the text.
7) Results of the quantitative analysis of phenolic compounds are shown in Table 2. As mentioned above, lignans were dominant phenolic compounds of the extract from S. chinensis fruits. The total lignan content was 29.61 ± 0.18 mg g−1 of the extract… mg g−1 should change to mg·g−1, Check and revise the entire text.
Response: We have corrected. Now, it is: “mg·g−1”.
- Figure 3. The results were considered significant at p<0.05 (*p<0.05, ns – not significant). When it comes to statistics, p needs to be italicized, please review and revise the entire text.
Response: We have corrected in the text of the manuscript. Now, it is: “p”.
- The author selected two extracts as positive controls, what I want to know is why not one chemical drug? Will the coagulation effect of plant extracts be better than that of chemical drugs? Please provide literature on these two extracts as positive controls to alleviate my concerns.
Response: We have added more information about it (the chapter of Introduction): “The properties of the extract from S. chinensis berries were compared to those of the extract from sea buckthorn (Hippophae rhamnoides L.) berries, and a commercial product – Aronox (Aronia melanocarpa berry extract), which have documented anti-platelet properties, and modulate other element of hemostasis [29–31; Cheng et al., 2003; Ryszawa et al., 2006; Luzak et al., 2010; Bijak et al., 2013; Sikora et al., 2012 and 2014; Shoatoor et al., 2021; Zhu et al., 2024].
10) loaded onto a short C18 column, 18 should be subscripted.
Response: We have corrected. Now, it is: “C18”.
11) Blood for the T-TAS assay was used within 2 hours after collection. 2 hours should change to 2 h. full blood collected in a tube with CDPA anticoagulant was centrifuged at 235xg for 12 minutes. 12 minutes should change to 12 min. 235xg should change to 235 xg. Check the entire text, scientific papers need to use international units instead of words.
Response: We have corrected. Now, it is: “12 min.”, “235 xg, and 2h”.
12) I have read all the references and found some issues. Refs 1, 3-4, Schisandra Chinensis needs to be italicized. Ref 48, FromSchisandra should change to From Schisandra. Genus and species names that involve Latin names need to be italicized. The author should review and revise all literature in accordance with the requirements of the journal.
Response: We have corrected.
13) The conclusion is consistent with the evidence and arguments provided. All the main questions raised by the author have been resolved.
Response: Thank you for your opinion.
14) The present study analyzed the berry extract of Schisandra chinensis, a dioecious vine known for its cardioprotective, antioxidant, and anti-inflammatory berries. The main component identified was dibenzocyclooctadiene lignans, with schisandrin being the dominant compound. The extract, at a concentration of 50 μg/mL, inhibited thrombus formation by 20%. Additionally, it reduced platelet adhesion to collagen across all tested concentrations (0.5-50 μg/mL) and to fibrinogen at higher concentrations (10-50 μg/mL). The extract also inhibited the activation of GPIIb/IIIa and P-selectin on platelets, while exhibiting no cytotoxicity. Although promising, further studies are required to elucidate its mechanisms of action and in vivo effectiveness.
Response: Our results indicate that S. chinensis berries could be used to produce natural nutrient supplements with cardioprotective potential, including anti-platelet activity. Nonetheless, the activity of S. chinensis still lacks validation in clinical settings. Future investigations into this extract and its chemical components, including schisandrin, should prioritize the following areas: (1) further understanding of the molecular mechanisms underlying its anti-platelet action, and the identification of specific targets; (2) bridging the gaps in knowledge about its in vivo efficiency by using animal models and performing clinical studies on healthy people and patients with different CVDs.
15) The other published materials on this topic mainly focus on other aspects of Schisandra chinensis, there is no attention paid to reports that berry extract of Schisandra chinensis as a promising therapeutic potential for the treatment of platelet aggregation. This study has added the above-mentioned blank. Moreover, a systematic study has been conducted.
Response: Thank you for your opinion.
16) The biggest problem with this study is many details need to be modified and improved. Too many issues can make people feel that the author's attitude is not rigorous.The author must take them seriously and make necessary revisions.
Response: Thank you for your helpful comments. All of them have been taken into consideration when revising the manuscript. We have added this information in text of manuscript (in red).
Round 2
Reviewer 1 Report (Previous Reviewer 1)
Comments and Suggestions for Authors
My concerns were addressed. One last point: the results of the T-TAS assay should be updated in the abstract (not 20% reduction of AUC10 but 15%)
Author Response
My concerns were addressed. One last point: the results of the T-TAS assay should be updated in the abstract (not 20% reduction of AUC10 but 15%)
Response: Thank you for your helpful comments. All of them have been taken into consideration when revising the manuscript. We have added this information in text of manuscript (in red).
We have corrected this result. Now, it is: “S. chinensis berry extract at the concentration of 50 μg/mL inhibited thrombus formation by approximately 15%.”
Reviewer 2 Report (New Reviewer)
Comments and Suggestions for Authors
The author revised the article according to my opinions and replied to my concerns. Therefore, I agree to accept this article after revising the following questions.
1) dry gas temperature 200 °C; There is no space between the number and °C, check and revise the full text.
2) The S. chinensis extract was separated on an ACQUITY UPLC BEH C18, 18 should subscript, check and revise the full text.
3) Then the platelet-rich plasma was collected and centrifuged at 1020xg for 15 min. 1020xg should change to 1020 xg.
Author Response
The author revised the article according to my opinions and replied to my concerns. Therefore, I agree to accept this article after revising the following questions.
Response: Thank you for your helpful comments. All of them have been taken into consideration when revising the manuscript. We have added this information in text of manuscript (in red).
- dry gas temperature 200 °C; There is no space between the number and °C, check and revise the full text.
Response: We have corrected. Now, it is: “dry gas temperature 200°C”.
- The chinensis extract was separated on an ACQUITY UPLC BEH C18, 18 should subscript, check and revise the full text.
Response: We have corrected. Now, it is: “C18”.
- Then the platelet-rich plasma was collected and centrifuged at 1020xg for 15 min. 1020xg should change to 1020 xg.
Response: We have corrected. Now, it is: “1020 xg”.
This manuscript is a resubmission of an earlier submission. The following is a list of the peer review reports and author responses from that submission.
Round 1
Reviewer 1 Report
Comments and Suggestions for Authors
In their work, Sławińska et al. evaluated the chemical composition and antiplatelet potential of the extracts from the Schisandra chinensis berries. The main findings reported in the study are: 1) schisandrin is the dominant compound among the phenolic constituents of the extracts, with other compounds present in smaller amounts; 2) the extract exerts antiplatelet activities by inhibiting platelet adhesion and activation with differential effects depending on the experimental conditions; 3) it reduces in vitro blood thrombus formation but does not alter routine coagulation tests like PT, aPTT and TT; 4) it is not cytotoxic to platelets. Although potentially interesting, the work presents several issues that need to be addressed:
Major points:
1. What is the extent of activation in resting platelets evaluated as exposure of active GPIIb-IIIa and P-sel? How much adhesion has been observed in resting platelets? By knowing the basal level of platelet activation and adhesion, the real magnitude of inhibition exerted by the extracts would be better appreciated. These control samples should be added to the graphs.
2. The results of the comparison between extracts from S. chinensis and the other berries are reported in a descriptive manner. However, they should be quantitative to appreciate the extent of the antiplatelet activity, if any, of all the comparators.
3. The Discussion section must be extensively revised. It reports unnecessary background information that would be more appropriately located in the Introduction section and should be shortened (e.g. page 12, lines 128-164 and page 13, lines 190-207). More importantly, it fails to adequately interpret and discuss the data. For example, how do authors harmonize the observations with isolated platelets with those in the global primary hemostasis assay? How would they describe the differential effect of the extract on platelet functions in relation to the experimental conditions (adhesive protein and agonist used)? How would they define the dose-effect relationship of the extract? What mechanisms do they propose for the observed effects? What concentrations of the extract are attainable upon administration to humans and therefore are pharmacologically relevant?
4. The protocol for platelet isolation should be described in the Methods section, as the reference cited is quite inaccurate in this respect. For example, did the authors prepare PRP by centrifuging blood at 1200g as reported in the reference? At this speed most of the platelets would sediment along with bigger cells (the required speed should not exceed 180-200g). Did the authors use platelet inhibitors during the washing steps (the reference does not report the use of any inhibitor)? In their absence platelets may be easily activated by the centrifugal force.
5. The method for studying primary hemostasis in whole blood with the TT Analysis System should be reported in greater detail, as it is not clear how primary hemostasis was specifically addressed, and the reference does not clarify the point. In particular, what is the PL-chip and how does it work?
Minor points:
1. Figures: what does the dotted line with asterisk mean? Probably that several concentrations are different from the control, but it should be clearly indicated in the captions. Moreover, there should be consistency between the figures, particularly regarding the use of horizontal dotted lines or multiple asterisks (e.g. compare Figg. 2A-B-D and 4C).
2. Page 7, line 21. The highest concentration of the extract inhibits adhesion of thrombin-activated platelets by roughly 30% (Fig.2B), and not 50% as reported. Please correct.
3. Page 12, lines 144-145. There are more than just two GPIIb-IIIa inhibitors, like abciximab. The reference (49) seems not adequate and should be replaced with a more comprehensive one (e.g. Sharifi-Rad, J., Sharopov, F., Ezzat, S.M. et al. An Updated Review on Glycoprotein IIb/IIIa Inhibitors as Antiplatelet Agents: Basic and Clinical Perspectives. High Blood Press Cardiovasc Prev. 2023;30:93-107).
4. Page 12, lines 146-147. Additional receptors, probably more important than GPIV, are present on the platelet surface and directly mediate the binding to collagen, like GPVI and GPIa-IIa (a2-b1).
5. Page 12, line 152. vWF is not a receptor.
6. Page 16, lines 355-356. Please specify that the samples were also incubated with the vehicle for the same time.
7. Page 17, line 419. The term “coagulometrically” means that a coagulometer was used, but what method does the instrument use? Turbidimetry? Magnetic beads? Please specify.
8. References identified as Olas, B. (2023a) and Olas, B. (2023b), number 35 and 36 of the reference list, are the same: Olas, B. (2023). Cardioprotective Potential of Berries of Schisandra chinensis Turcz. (Baill.), Their Components and Food Products. Nutrients, 15(3), 592. https://doi.org/10.3390/nu15030592.
Author Response
In their work, Sławińska et al. evaluated the chemical composition and antiplatelet potential of the extracts from the Schisandra chinensis berries. The main findings reported in the study are: 1) schisandrin is the dominant compound among the phenolic constituents of the extracts, with other compounds present in smaller amounts; 2) the extract exerts antiplatelet activities by inhibiting platelet adhesion and activation with differential effects depending on the experimental conditions; 3) it reduces in vitro blood thrombus formation but does not alter routine coagulation tests like PT, aPTT and TT; 4) it is not cytotoxic to platelets. Although potentially interesting, the work presents several issues that need to be addressed:
Thank you for reviewing the manuscript and providing such helpful comments. All of them have been taken into consideration when revising the manuscript.
Major points:
- What is the extent of activation in resting platelets evaluated as exposure of active GPIIb-IIIa and P-sel? How much adhesion has been observed in resting platelets? By knowing the basal level of platelet activation and adhesion, the real magnitude of inhibition exerted by the extracts would be better appreciated. These control samples should be added to the graphs.
Response: We added negative control samples to the graphs.
- The results of the comparison between extracts from S. chinensis and the other berries are reported in a descriptive manner. However, they should be quantitative to appreciate the extent of the antiplatelet activity, if any, of all the comparators.
Response: We have added quantitative analysis in Table 3.
- The Discussion section must be extensively revised. It reports unnecessary background information that would be more appropriately located in the Introduction section and should be shortened (e.g. page 12, lines 128-164 and page 13, lines 190-207). More importantly, it fails to adequately interpret and discuss the data. For example, how do authors harmonize the observations with isolated platelets with those in the global primary hemostasis assay? How would they describe the differential effect of the extract on platelet functions in relation to the experimental conditions (adhesive protein and agonist used)? How would they define the dose-effect relationship of the extract? What mechanisms do they propose for the observed effects? What concentrations of the extract are attainable upon administration to humans and therefore are pharmacologically relevant?
Response: We have changed Introduction, and moved some of the information from Discussion there. We have also modified and added new information to the Discussion section, for example:
“The concentrations used in our study (0.5-50 μg/mL) are generally attainable upon oral administration of plant extracts to humans [51–53]. Only the highest concentration of the extract (50 μg/mL) had statistically significant effect on the exposition of the active form of GPIIb/IIIa and thrombus formation in full blood, however in most of the other assays lower concentrations (0.5-10 μg/mL) had significant antiplatelet effects as well.”
“Platelet isolation protocol can strongly influence the platelet response. Ideally, all assays should be performed in conditions that are as close to the natural platelet environment as possible, however some protocols require the separation of platelets from other blood components [44]. Using both methods allows for broader understanding of platelet function. Here, the extract from S. chinensis showed antiplatelet activity in both methods that utilized full blood (flow cytometry and T‑TAS) and washed platelets (platelet adhesion to fibrinogen and collagen).“
“however not much is known about its mechanisms of action. In a study by Jung et al. (1997), schisandrin A and B significantly inhibited the binding of platelet activating factor (PAF) to rabbit blood platelets [17]. This could explain the antiplatelet activity of S. chinensis extract, however more studies that explore its potential mechanisms of action needed to fully understand its properties.”
“The role of GPIIb/IIIa is to facilitate platelet aggregation through fibrinogen binding; ADP mediates this process through platelet activation [24]. The extract from S. chinensis inhibited both ADP-induced exposition of GPIIb/IIIa on the surface of platelets and adhesion of ADP-stimulated (but not thrombin-stimulated) platelets to fibrinogen. Interestingly, ADP stimulation had no effect on the exposition of P-selectin on the platelet surface. This suggests that S. chinensis might interfere with the process of ADP-induced change of conformation of GPIIb/IIIa from its low-affinity to high-affinity state or fibrinogen binding. This could be achieved through interaction with different signaling molecules. For example, Ginsenoside-Rp3 (a Ginseng saponin) decreased ADP-induced platelet aggregation, fibrinogen binding, and fibronectin adhesion to GPIIb/IIIa through the inhibition of Src family kinases (SFKs), Src-dependent phospholipase Cγ2 (PLCγ2), and phosphatidylinositol 3-kinase/Akt (PI3K/Akt) signaling pathways [51].
- The protocol for platelet isolation should be described in the Methods section, as the reference cited is quite inaccurate in this respect. For example, did the authors prepare PRP by centrifuging blood at 1200g as reported in the reference? At this speed most of the platelets would sediment along with bigger cells (the required speed should not exceed 180-200g). Did the authors use platelet inhibitors during the washing steps (the reference does not report the use of any inhibitor)? In their absence platelets may be easily activated by the centrifugal force.
Response: We added the protocol for platelets isolation in the methods section. Full blood was centrifuged at 235g (1200 rpm) to obtain PRP. We did not use inhibitors, if the platelets were activated we did not include that sample in the analysis and repeated the isolation on another blood sample. Platelet isolation without inhibitors was also used in other studies, for example in (Truss et al. (2009) (doi: 10.1111/j.1538-7836.2009.03589.x), Rodríguez et al. (2024) doi: (10.1016/j.biopha.2024.117154), Morihara and Hino (2017) (doi: 10.1007/s11418-016-1055-4), and Son et al. (2017) (doi: 10.1186/s12906-017-2032-5).
- The method for studying primary hemostasis in whole blood with the TT Analysis System should be reported in greater detail, as it is not clear how primary hemostasis was specifically addressed, and the reference does not clarify the point. In particular, what is the PL-chip and how does it work?
Response: We have added information about T-TAS and PL-Chip in the methods section. The T-TAS is a standardized whole blood flow chamber system for the measurement of in-vitro thrombus formation in ready to use pre-coated chips with microcapillary flow channels. The system includes a platelet chip (PL-chip), which contains 26 microcapillaries coated with type I collagen and shear rates of 1500/s.
T-TAS (Fujimori Kogyo Co. Ltd, ZACROS, Japan) was performed according to the manufacturer's instruction. For the T-TAS measurements, blood was pipetted into the Reservoir Set attached to the PL-chip’s flow path. In the PL-chip, BAPA anticoagulated blood is pumped through collagen-coated microcapillaries with a 1500/s wall shear rate which causes platelet activation and formation of platelet thrombi. The newly-formed thrombi block the flow path, which increases the pressure inside the cappilaries. The BAPA anticoagulant blocks secondary hemostasis through the inhibition of thrombin and factor Xa, so only primary hemostasis can take place.
Minor points:
- Figures: what does the dotted line with asterisk mean? Probably that several concentrations are different from the control, but it should be clearly indicated in the captions. Moreover, there should be consistency between the figures, particularly regarding the use of horizontal dotted lines or multiple asterisks (e.g. compare Figg. 2A-B-D and 4C).
Response: We corrected this – for the sake of clarity we added asterisks above every concentration instead of using dotted lines.
- Page 7, line 21. The highest concentration of the extract inhibits adhesion of thrombin-activated platelets by roughly 30% (Fig.2B), and not 50% as reported. Please correct.
Response: We have corrected this. Now, it is: “at the highest concentration (50 µg/mL), the extract inhibited the adhesion of thrombin-activated blood platelets by about 30% in comparison with control.”
- Page 12, lines 144-145. There are more than just two GPIIb-IIIa inhibitors, like abciximab. The reference (49) seems not adequate and should be replaced with a more comprehensive one (e.g. Sharifi-Rad, J., Sharopov, F., Ezzat, S.M. et al. An Updated Review on Glycoprotein IIb/IIIa Inhibitors as Antiplatelet Agents: Basic and Clinical Perspectives. High Blood Press Cardiovasc Prev. 2023;30:93-107).
Response: We have added more information about it: “Currently, there are three GP IIb/IIIa inhibitors (tirofiban, eptifibatide, and abciximab) used as anti-platelet drugs (Sharaifi-Rad et al., 2023; Tummala & Rai, 2024).”
- Page 12, lines 146-147. Additional receptors, probably more important than GPIV, are present on the platelet surface and directly mediate the binding to collagen, like GPVI and GPIa-IIa (a2-b1).
Response: We have added more information about it: “Platelet adhesion is mediated mainly by GPIb-IX-V (which binds to von Willebrand factor (vWF), which in turn binds to collagen) and GPVI, GPIa/IIa, or GPIV, which bind to collagen directly [22–25].”
- Page 12, line 152. vWF is not a receptor.
Response: We corrected this.
- Page 16, lines 355-356. Please specify that the samples were also incubated with the vehicle for the same time.
Response: We added that information.
- Page 17, line 419. The term “coagulometrically” means that a coagulometer was used, but what method does the instrument use? Turbidimetry? Magnetic beads? Please specify.
Response: We added this information “(optical method based on measurements of turbidity)”.
- References identified as Olas, B. (2023a) and Olas, B. (2023b), number 35 and 36 of the reference list, are the same: Olas, B. (2023). Cardioprotective Potential of Berries of Schisandra chinensis Turcz. (Baill.), Their Components and Food Products. Nutrients, 15(3), 592. https://doi.org/10.3390/nu15030592.
Response: We corrected this.
Reviewer 2 Report
Comments and Suggestions for Authors
In their manuscript, Sławińska et al. analyse extracts from berries of Schisandra chinensis on platelet function, thrombus formation, and coagulation. Plant-derived nutrition supplements are widely used; however, careful examinations of active compounds' effects are needed. Experimental settings, data presentations, and conclusions provided in the manuscript do not clearly determine the use of extracts.
In the Results, mention the method used to detect extracted compounds. It would be also valuable to have an explanation of the compound groups (lignan, flavonoid anthocyanidins) in the beginning, e.g. it is difficult to follow the description in the text and Table 2.
Figure 2. If platelets adhere to some of the mentioned coated surfaces (collagen, fibrinogen), then they are activated, not resting. There is no clear concentration curve of the extract effects on adhesion in 2A and 2B. Statistical significance should be shown for each condition (also for all other figures in the manuscript). To verify the results obtained measuring enzyme activity, experiments should be performed by immunofluorescence (with e.g. actin staining to monitor platelet adhesion), and images of adhered platelets should be shown at least for some conditions (control, high extract concentrations).
Figures 3 and 4. If only 12-15% for PAC1, and 25-30% for CD62P of platelets is positive after ADP stimulation, this means that platelet activation is weak and there might be something wrong with the experimental setting. Collagen cannot be used in flow cytometry, because it is a fiber that binds to and aggregates platelets, therefore platelets are not represented anymore as single events in the same gate in flow cytometry. Convulxin or CRP (that activates single receptors on platelets) can be used instead of collagen. Experiments should be established on washed platelets first. Also, authors should show if treatment with extract (different concentrations) affects platelets (PAC1, CD62P).
Results on LDH should be shown. Other toxicity tests more specific to platelets would be more informative (mitochondria potential).
Figure 5. It would be worth analyzing, or commenting on time kinetics in thrombus formation, in Results. Also, is platelet aggregation in whole blood or platelet-rich plasma affected with extracts?
Table 3. It is necessary to include comparison results (other plant extracts, Aronia, Sea buckthorn) in previous graphs (experiments performed in parallel with examined S. chinensis extracts).
Author Response
In their manuscript, Sławińska et al. analyse extracts from berries of Schisandra chinensis on platelet function, thrombus formation, and coagulation. Plant-derived nutrition supplements are widely used; however, careful examinations of active compounds' effects are needed. Experimental settings, data presentations, and conclusions provided in the manuscript do not clearly determine the use of extracts.
Thank you for reviewing the manuscript and providing such helpful comments. All of them have been taken into consideration when revising the manuscript.
In the Results, mention the method used to detect extracted compounds. It would be also valuable to have an explanation of the compound groups (lignan, flavonoid anthocyanidins) in the beginning, e.g. it is difficult to follow the description in the text and Table 2.
Response: The additional descriptions you suggested have been added.
Figure 2. If platelets adhere to some of the mentioned coated surfaces (collagen, fibrinogen), then they are activated, not resting. There is no clear concentration curve of the extract effects on adhesion in 2A and 2B. Statistical significance should be shown for each condition (also for all other figures in the manuscript). To verify the results obtained measuring enzyme activity, experiments should be performed by immunofluorescence (with e.g. actin staining to monitor platelet adhesion), and images of adhered platelets should be shown at least for some conditions (control, high extract concentrations).
Response: We replaced “resting” with “unstimulated”. For not significant results, we added “ns” to the graphs, while statistically significant results are indicated by asterisks.
In the earlier and the present experiments we observed that anti-platelet activity of tested plant extracts and bioactive compounds isolated from these extracts do not always depend on the concentration. Other authors (for example, Wang et al., Sci Reports, 2020) also noted that the effect of schisandrin B on various biological functions of cells is not always concentration-dependent.
We decided not to use immunofluorescence, because we used the method for measuring enzyme activity that was described by (Bellavite, P.; Andrioli, G.; Guzzo, P.; Arigliano, P.; Chirumbolo, S.; Manzato, F.; Santonastaso, C. A Colorimetric Method for the Measurement of Platelet Adhesion in Microtiter Plates. Anal. Biochem. 1994, 216, 444–450); this method was also used in other publications measuring blood platelet adhesion (Truss et al., 2009; Braune-Zhou et al., 2015, and other). For example, Braune-Zhou (2015) observed that a colorimetric assay (acid phosphatase (ACP)) allowed a similar accurate quantification of the platelet adhesion compared to the microscopic evaluation.
Figures 3 and 4. If only 12-15% for PAC1, and 25-30% for CD62P of platelets is positive after ADP stimulation, this means that platelet activation is weak and there might be something wrong with the experimental setting. Collagen cannot be used in flow cytometry, because it is a fiber that binds to and aggregates platelets, therefore platelets are not represented anymore as single events in the same gate in flow cytometry. Convulxin or CRP (that activates single receptors on platelets) can be used instead of collagen. Experiments should be established on washed platelets first. Also, authors should show if treatment with extract (different concentrations) affects platelets (PAC1, CD62P).
Response: We have used the two concentrations of ADP (10 and 20 µM) and one concentration of collagen (10 µg/mL), because they were also used in WataÅ‚a's publications and other publications that studied blood platelets. That’s why we decided to use the same agonists in our studies. For example, Roberts et al. (J. Biol. Chem. 2003), Rywaniak et al. (Platelets 2014), TomczyÅ„ska et al. (J. Cell Mol. Med. 2018), Rofriguez et al. (BP 2024), and Morel et al. (Mol. Cell Biochem. 2017) have used these agonists, including collagen. In addition, results of these publications (for example, Morel et al. (Mol. Cell Biochem. 2017) and TomczyÅ„ska et al. (J. Cell Mol. Med. 2018) for PAC1 and CD62P of platelets is positive after ADP stimulation are very similar for our results. For example, for PAC1 of platelets (after 20 µM ADP)– about 8 [%] (Morel et al. (Mol. Cell Biochem. 2017), for PAC1 of platelets (after 20 µM ADP)– about 12 [%] (Fig. 3B); for CD62P of platelets (after 20 µM ADP)– about 20 [%] (Morel et al. (Mol. Cell Biochem. 2017), for CD62P of platelets (after 20 µM ADP)– about 32 [%] (Fig. 4B).
We have presented the results in the Results section:
Effect of the extract from S. chinensis berries on the adhesion of washed blood platelets to collagen and fibrinogen
Effect of the extract from S. chinensis berries on parameters of blood platelet activation measured with flow cytometry in whole blood
Results on LDH should be shown. Other toxicity tests more specific to platelets would be more informative (mitochondria potential).
Response: We added a graph depicting the effect of the extract on LDH activity.
Figure 5. It would be worth analyzing, or commenting on time kinetics in thrombus formation, in Results. Also, is platelet aggregation in whole blood or platelet-rich plasma affected with extracts?
Response: We added the charts depicting changes in pressure in the PL-chip (Figure 5B) and commented on it. “Figure 5B shows changes in pressure that were recorded within the PL-chip in each sample at the highest concentration of the extract (50 μg/mL). The samples differed between each other, however in 4 out of 6 samples the antithrombotic activity of the extract took effect in the later stages of the test. At first the pressure was similar in the control and the extract-containing sample; after a while maximum pressure was recorded in the control sample, while the extract prolonged the time needed to reach total occlusion.” We did not measure platelet aggregation in this study; however, in a study by Kim et al. (2017) The mix of extracts from S. chinensis and Morus alba inhibited collagen-stimulated platelet aggregation measured by light transmission aggregometry.
Table 3. It is necessary to include comparison results (other plant extracts, Aronia, Sea buckthorn) in previous graphs (experiments performed in parallel with examined S. chinensis extracts).
Response: We have added quantitative analysis in Table 3.
Round 2
Reviewer 1 Report
Comments and Suggestions for Authors
The Authors addressed my concerns. Only a few minor points remain.
1. Page 14, lines 154-156. I think that this point deserves further discussion. The prolongation of time to “thrombus” formation was observed only with the highest extract concentration and was small. However, the antithrombotic activity of the extract could have been underestimated under the experimental conditions employed. In fact, the design of the assay excludes the contribution of thrombin to the formation of the platelet plug by using BAPA, and therefore one of the targets for the antiplatelet activity of the extract (which inhibits thrombin-induced platelet adhesion to collagen at all concentrations) is overlooked. Moreover, other functions of thrombin on platelets could potentially be inhibited.
2. In the discussion, lines 172-173, the Authors should acknowledge that PT, aPTT and TT are screening assays with technical limitations, and that more sensitive and comprehensive assays, e.g. thrombin generation, are needed to confirm the absence of any effect on the clotting pathways.
3. I would substitute the word “thrombus” referred to the results with the T-TAS (PL-chip) assay with “platelet plug” throughout the text, as in the absence of thrombin-mediated fibrin formation only the platelet component is present.
4. Figure 5B. This new panel depicts the pressure variations in each sample analyzed. As in two out of six samples (2 and especially 5) the pressure curve rises faster in the presence of the extract, and the tracings are inherently complex, it would be advisable to show also the individual AUC10 values.
5. Table 3. In the same way non-significant values were reported in case of “no effect”, the same should be done for the “increase” results. % variation would be presumably a negative value, which should be explained in the caption.
6. Page 9, caption to Fig. 3, line 69. The p value of the difference between controls in B is missing.
7. Pag.14, line 182: are needed
Author Response
The Authors addressed my concerns. Only a few minor points remain.
Thank you for reviewing the manuscript and providing such helpful comments. All of them have been taken into consideration when revising the manuscript.
- Page 14, lines 154-156. I think that this point deserves further discussion. The prolongation of time to “thrombus” formation was observed only with the highest extract concentration and was small. However, the antithrombotic activity of the extract could have been underestimated under the experimental conditions employed. In fact, the design of the assay excludes the contribution of thrombin to the formation of the platelet plug by using BAPA, and therefore one of the targets for the antiplatelet activity of the extract (which inhibits thrombin-induced platelet adhesion to collagen at all concentrations) is overlooked. Moreover, other functions of thrombin on platelets could potentially be inhibited.
Response: We added this point to Discussion “This effect was observed only at the highest concentration and was quite small, however it is important to note that the antithrombotic activity of the extract could have been underestimated due to the character of the assay. The BAPA anticoagulant employed in the PL-chip blocks secondary hemostasis, including thrombin activity. The fact that the extract significantly inhibited the adhesion of thrombin-activated platelets to collagen at all used concentrations suggests that thrombin might be one of the targets of its antiplatelet activity.”
- In the discussion, lines 172-173, the Authors should acknowledge that PT, aPTT and TT are screening assays with technical limitations, and that more sensitive and comprehensive assays, e.g. thrombin generation, are needed to confirm the absence of any effect on the clotting pathways.
Response: We have added this information: “It is important to note that PT, APTT and TT are screening assays with technical limitations, and more sensitive and comprehensive assays, e.g. thrombin generation, are needed to confirm the absence of any effect on the clotting pathways.”.
- I would substitute the word “thrombus” referred to the results with the T-TAS (PL-chip) assay with “platelet plug” throughout the text, as in the absence of thrombin-mediated fibrin formation only the platelet component is present.
Response: We have changed this. Now, it is: “platelet plug”.
- Figure 5B. This new panel depicts the pressure variations in each sample analyzed. As in two out of six samples (2 and especially 5) the pressure curve rises faster in the presence of the extract, and the tracings are inherently complex, it would be advisable to show also the individual AUC10 values.
Response: We added the AUC10 values to Figure 5B.
- Table 3. In the same way non-significant values were reported in case of “no effect”, the same should be done for the “increase” results. % variation would be presumably a negative value, which should be explained in the caption.
Response: We have changed Table 3. We have added the “increase” results. For example, “No inhibitory effect (only increase; 12.4±5.9; p>0.05)”.
- Page 9, caption to Fig. 3, line 69. The p value of the difference between controls in B is missing.
Response: We have added this information. Now, it is: “The differences between controls were significant (p<0.01 (A), p<0.05 (B, C)).”.
- Pag.14, line 182: are needed.
Response: We have corrected this. Now, it is: “Two other compounds from S. chinensis fruits (pregomisin and chamigrenal) also showed antagonistic activity toward PAF, though their activity was relatively weak”.